

# Reaction between perfluoroaldehydes and hydroperoxy radical in the atmosphere: Reaction mechanisms, reaction kinetics modelling, and atmospheric implications

Zegang Dong[1,2], Chaolu Xie[3], Bo Long[1]

[1]School of Materials Science and Engineering, Guizhou Minzu University, Guiyang, 550025, China
[2]School of Chemistry and Chemistry Engineering, South China University of Technology, Guangzhou, 510006, China
[3]College of Physics and Mechatronic Engineering, Guizhou Minzu University, Guiyang, 550025, China

*Correspondence to:* Bo Long (wwwltcommon@sina.com)

**ABSTRACT:** Linear perfluoroaldehydes are important products formed in the atmospheric oxidation of industrial fluorinated
compounds. However, their atmospheric lifetimes are incompletely known. Here, we employ high level quantum chemistry
methods and a dual-level strategy for kinetics to probe the reactions of $C_2F_5CHO$ and $C_3F_7CHO$ with $HO_2$. Our calculated
results unveil almost equal activation enthalpies at 0 K for perfluoroaldehyde reaction with $HO_2$, indicating that the carbon
chain length minimally influences reaction thermodynamics. Interestingly, the present findings reveal that anharmonicity
remarkably enhances the reaction rate constant, whereas multi-structural anharmonicity, recrossing, and tunnelling effects
exhibit lesser impacts in the $C_2F_5CHO/C_3F_7CHO + HO_2$ reaction. In particular, the atmospheric lifetimes for $C_2F_5CHO$ and
$C_3F_7CHO$, approximately 14.4-31.3 hours and 21.6-51.8 hours by $HO_2$ are much shorter than those via OH radical,
underscoring the dominant removal role of $HO_2$ toward $C_2F_5CHO$ and $C_3F_7CHO$ in the atmosphere. Since GEOS-Chem
simulation shows that the concentration of $HO_2$ is at least $10^2$ times higher than that of OH radical in Russia, Malaysia, and
parts of Africa, the reactions of $C_2F_5CHO$ and $C_3F_7CHO$ with $HO_2$ radicals dominate over those with OH radicals and play
more vital role in the atmospheric chemical processes of these regions. This study enhances our understanding of the chemical
transformations of linear perfluoroaldehydes and provides a scientific foundation for strategies aimed at mitigating their
emissions.

## 1. Introduction

Poly- and perfluoroalkyl substances (PFASs) are highly fluorinated compounds with long atmospheric lifetimes, which have
important influences on global warming potential (GWP) and environmental health.(Ackerman Grunfeld et al., 2024; Rupp et
al., 2023; Sznajder-Katarzyńska et al., 2019; Wu et al., 2024) Linear perfluoroaldehydes ($C_nF_{2n+1}CHO$) are significant
intermediate compounds, which belong to the aldehydes family of PFASs.(Li et al., 2024; Thackray et al., 2020)
Chlorofluorocarbons (CFCs) and their temporary replacements, hydrochlorofluorocarbons (HCFCs), hydrofluorocarbons
(HFCs), and hydrofluoroolefins (HFOs) are the important source of the linear perfluoroaldehydes.(Burkholder et al., 2015;



Hurley et al., 2006; Martin et al., 2005; Rand and Mabury, 2017; Wang et al., 2023; Waterland and Dobbs, 2007) For example, under low $NO_x$ conditions, the reaction of OH radicals with the potential foaming agent $CF_3(CF_2)_2CH=CH_2$ (HFC-1447fz) leads to the formation of $C_3F_7CHO$.(Jiménez et al., 2016; Yu et al., 2024) Furthermore, the atmospheric chemical processes of linear perfluoroaldehydes are of key importance for determining the atmospheric oxidation of fluorotelomer alcohols (FTOHs).(Antiñolo et al., 2012; Hurley et al., 2004)

Linear perfluoroaldehydes were generally considered to be removed through free radical reactions (such as Cl, OH·, etc.)(Andersen et al., 2004; Chiappero et al., 2010; Wang et al., 2007) and photochemical reactions.(Chiappero et al., 2006; Kelly et al., 2004) During the daytime, photolysis of linear perfluoroaldehydes was considered to be the dominant removal process for $C_nF_{2n+1}CHO$, with estimated atmospheric lifetimes ranging from hours to several days.(Antiñolo et al., 2014; Chiappero et al., 2006) Antiñolo et al. (2014) reported that the photolysis lifetime of $C_2F_5CHO$ is expected to be 3.5 hours at

273 K, with the main degradation products of $CF_3CFO$ and $COF_2$. In addition, studies have shown that the length of the carbon chain in $C_nF_{2n+1}CHO$ significantly affects the quantum yield of photolysis.(Chiappero et al., 2006; Kelly et al., 2004) During the nighttime, the reactions of free radicals with $C_nF_{2n+1}CHO$ were considered to be the major degradation pathways. However, previous studies reported relatively slow rate constants for the reaction between OH and $C_nF_{2n+1}CHO$ (n=1-4) with the values of $(6.5 \pm 1.2) \times 10^{-13}$, $(5.57 \pm 0.07) \times 10^{-13}$, $(5.8 \pm 0.6) \times 10^{-13}$, and $(6.1 \pm 0.5) \times 10^{-13}$ $cm^3$ molecule$^{-1}$ s$^{-1}$, respectively, at 298

K.(Andersen et al., 2004; Antiñolo et al., 2014; Solignac et al., 2007) This corresponds to a longer atmospheric lifetime > 20 days for these linear perfluoroaldehydes. Moreover, the rate constant of Cl atoms with $C_nF_{2n+1}CHO$ (1,2, 3, 4) is around $2 \times 10^{-12}$ $cm^3$ molecule$^{-1}$ s$^{-1}$, which is slightly faster than that of the OH radical reactions.(Andersen et al., 2004; Sulbaek Andersen et al., 2003) The long atmospheric lifetimes of $C_nF_{2n+1}CHO$ provide an opportunity for other atmospheric oxidation processes of $C_nF_{2n+1}CHO$ by other atmospheric oxidants.

$HO_2$ radicals are of ubiquitous active species in the atmosphere with the concentration being two orders of magnitude higher than that of OH radicals.(Bottorff et al., 2023; Gao et al., 2024a; Sascha et al., 2019; Zhang et al., 2019, 2024, 2022) Moreover, previous investigations have shown that the reactions of aldehydes with $HO_2$ affect the degradation process of aldehydes.(Hermans et al., 2005; Sascha et al., 2019; Zhou et al., 2024) Additionally, global three-dimensional chemistry-transport model calculations suggest that the oxidation reactions of formaldehyde and acetone initiated by hydroperoxyl radical

contribute to 30% loss of formaldehyde and acetone at the tropical troposphere.(Hermans et al., 2005) Nevertheless, the importance of sink pathway by $HO_2$ is still unknown because there have not been kinetics data for linear perfluoroaldehydes with $HO_2$ in the literature.

Here, we have investigated the reactions of $HO_2$ with linear perfluoroaldehydes $C_nF_{2n+1}CHO$ (n = 2-5), specifically focusing on $C_2F_5CHO$ and $C_3F_7CHO$, referred to as reactions (R1) and (R2) respectively. To delve into these reactions, high-level

quantum chemistry calculation close to CCSDT(Q) accuracy in conjunction with dual-level strategy were performed to obtain their quantitative kinetics. Simultaneously, to provide further insight into kinetics, we detailedly evaluate the impact of various parameters, including torsional anharmonicity, anharmonicity on the reaction kinetics over atmosphere-related temperatures and pressures. In addition, the chemical transformation of the formed intermediate products has been discussed in reactions



R1 and R2. We further estimate the enthalpies of activation at 0 K for the larger-sized reactions of longer-chain perfluoroaldehyde with HO$_2$. Moreover, we also discuss the importance of these reactions R1 and R2 by combining the calculated reaction kinetics with global atmospheric modelling. The current results not only provide a comparative analysis with the kinetics of analogous OH-initiated reactions and photolytic processes, but also extend our understanding of the role of HO$_2$ in modulating the atmospheric lifetime of perfluoroaldehydes. These findings could hold significant implications for the formation and yield of HCOOH, thereby contributing to the broader discourse on atmospheric chemistry and environmental policy.

$$C_2F_5CHO + HO_2 \rightarrow C_2F_5CH(OH)OO \tag{R1}$$

$$C_3F_7CHO + HO_2 \rightarrow C_3F_7CH(OH)OO \tag{R2}$$

## 2. Computational methods and atmospheric modelling

### 2.1 Options for electronic structure density functionals

Our goal is to establish a precise set of electronic structure and kinetic calculation methods for the XCHO + HO$_2$ reaction, delivering satisfactory quantitative results.(Long et al., 2022) This previous study indicated that the CCSD(T)-F12a/cc-pVTZ-F12//M06-2X/MG3S theoretical methods can make good agreement with beyond-CCSD(T) results for the similar reaction of HCHO + HO$_2$.(Long et al., 2022) Consequently, we intend to utilize the well-validated methods in the present investigations in reactions R1 and R2. Specifically, the M06-2X(Zhao and Truhlar, 2008b, a) density functional with the MG3S(Lynch et al., 2003) basis set was employed to optimize the geometries, while CCSD(T)-F12a(Adler et al., 2007; Knizia et al., 2009)/cc-pVTZ-F12 for R1 and R2 and FNO-CCSD(T)-F12(Gyevi-Nagy et al., 2021; Taube and Bartlett, 2008)/cc-pVDZ-F12 for other C$_n$F$_{2n+1}$CHO + HO$_2$ (n = 1 - 5) were used to calculate single-point energies. Furthermore, intrinsic reaction coordinate (IRC) calculation was done to determine the correct transition states by examining the connections of each saddle point to its corresponding minima.(Hratchian and Schlegel, 2004, 2005; Kenyon, 1968)

### 2.2 Vibrational frequencies

We found that standard scale factor is actually not applicable for some transition states in previous investigation, so we used two scale factors.(Zheng et al., 2014, 2015) The standard scale factor for M06-2X/MG3S is 0.970. Furthermore, we also calculated the specific reaction scale factors to assess the effects of anharmonicity. The reaction-specific scale factors were obtained by using the MPW1K/6-311+G(2df, 2p) electronic structure method based on the hybrid degeneracy-corrected second-order vibrational perturbation theory (HDCVPT).(Bloino et al., 2012; Kuhler et al., 1996) This is necessary and effective for eliminating the activation enthalpy error caused by the standard scale factors and the results were list in Table S1 and Table S2,Supplement. This was obtained by equation (1),

$$\lambda^{SRP,ZPE} = \lambda^{Anh}\lambda^{H} \tag{1}$$



where $\lambda^{Anh}$ is the ratio of anharmonic zero-point vibrational energies (ZPE) to harmonic ZPE at the MPW1K/6-311+G(2df,

2p) level. $\lambda^H$ is 0.983 for M06-2X/MG3S to correct harmonic frequencies. The result shows that the specific reaction scale factors are 0.955 for TS1 (See Figure 1) and 0.956 for TS2 (See Figure 1), which is a large deviation from the standard value of 0.970; this results in a decrease in calculated enthalpies of activation of 0.72 and 0.78 for TS1 and TS2 at 0 K, respectively. In addition, multi-structural torsional anharmonicity involving reactant and transition state were all calculated using MS-T method (multi-structural method for torsional anharmonicity).(Yu et al., 2012; Zheng et al., 2011; Zheng and Truhlar, 2013)

## 2.3 kinetics calculations

The dual-level strategy was utilized to compute the high-pressure limit rate constant.(Long et al., 2016, 2019; Xia et al., 2023) As shown in equation (2), we integrated a conventional transition-state theory rate constant $k_{TST}^{HL}$ predicated on higher-level (HL, CCSD(T)-F12a/cc-pVTZ-F12//M06-2X/MG3S) inputs with transmission coefficients derived from direct dynamics at a lower level (LL,M11-L/MG3S), employing a specific density functional that is chose from the results of benchmark

calculations (see Table S3). We have incorporated both a recrossing transmission coefficient $\Gamma_{CVT/TST}^{LL}$ and a tunneling transmission coefficient $k_{SCT}^{LL}$, as calculated through reaction-path variational transition state theory, with a particular emphasis on the canonical variational theory coupled with small-curvature tunneling (CVT/SCT).(Garrett and Truhlar, 1979; Liu et al., 1993; Truhlar et al., 1982) Additionally, a multi-structural transmission coefficient ($F^{MS-T}$) was introduced to this framework to cancel the errors caused by the multi-structural anharmonicity, thereby advancing our approach to the DL-MS-CVT/SCT

method, which provides a detailed and multifaceted treatment of the rate constant calculation, and can effectively obtain quantitative kinetics.

$$k_{MS-CVT/SCT}^{DL} = F^{MS-T} \times k_{TST}^{HL} \times k_{SCT}^{LL} \times \Gamma_{CVT/TST}^{LL} \tag{2}$$

The pressure-dependent rate constants were done by employing the system-specific quantum Rice-Ramsperger-Kassel (SS-QRRK) theory in the temperature range of 190-350 K.(Bao et al., 2016b, a; Bao and Truhlar, 2017) This method relies only

on the high-pressure limiting rate constant that was calculated by the dual-level strategy. The computational details of pressure-dependent rate constants are presented in the Supplement.

## 2.4 Atmospheric modelling

We used GEOS-Chem 14.4.2 with a horizontal resolution of $2.0° \times 2.5°$ to simulate space distribution of $HO_2$ and OH at 47 vertical layers in the period from February 2018 to February 2019.(Bey et al., 2001) The time includes six months of spin-up

and output per hour. GEOS-Chem is a global, three-dimensional chemical transport model associated with atmospheric composition (http://geos-chem.org). Modern-Era Retrospective analysis for Research and Applications, Version 2 (MERRA-2)(Gelaro et al., 2017) was used as meteorological field data and Harmonized Emissions Component (HEMCO 3.9) was used as the source of emissions data.(Lin et al., 2021) The emissions include biogenic emissions from Model of Emissions of Gases and Aerosols from Nature (MEGANv2.1)(Hu et al., 2015; McDuffie et al., 2020) and anthropogenic emissions from the global



Community Emissions Data System (CEDS) (McDuffie et al., 2020) inventory. Simulation uses default full chemistry mechanism including $HO_x$-$NO_x$-VOC-$O_3$-halogen chemistry, which is done by our previous investigation.(Bloss et al., 2007)

**2.5 Software**

All density function calculation, including Zero-point energy (ZPE) correction were carried out using Gaussian 16 software package,(Zhao and Truhlar, 2008b) and the single point energy calculations for CCSD(T)-F12a/cc-pVTZ-F12 and FNO-

CCSD(T)-F12a/cc-pVDZ-F12 were done using Molpro 2019(Werner et al., 2019) and MRCC code.(Kállay et al., 2020; Mihály Kállay, P. Nagy, Dávid Mester, Z. Rolik, Gyula Samu, J. Csontos, József Csóka, P. Szabó, László Gyevi-Nagy, Bence Hégely, István Ladjánszki, Lóránt Szegedy, Bence Ladóczki, K. Petrov, M. Farkas, Pál D. Mezei, n.d.) MS-T method was executed through MSTor-2023 program package.(Chen et al., 2023) The rate constants were done with the KiSThelP,(Canneaux et al., 2014) Polyrate 2017-C, and Gaussrate 2017-B.(Zheng et al., 2017, n.d.)

**3. Results and discussion**

**3.1. The electronic structure of the $C_2F_5CHO$/$C_3F_7CHO$ + $HO_2$ reaction**

We considered the $C_2F_5CHO$/$C_3F_7CHO$ + $HO_2$ reaction similar to the reactions of aldehydes with $HO_2$.(Long et al., 2022) The dominant mechanism is that the hydrogen atom of $HO_2$ is transferred to the terminal oxygen atom of $C_2F_5CHO$/$C_3F_7CHO$, and simultaneously, the oxygen atom of $HO_2$ is connected to carbon atom of carbonyl group of $C_2F_5CHO$/$C_3F_7CHO$. Figure 1

depicts ZPE corrected potential energy profile of the reaction of $C_2F_5CHO$/$C_3F_7CHO$ + $HO_2$ at the CCSD(T)-F12a/cc-pVTZ-F12//M06-2X/MG3S level. $C_2F_5CHO$/$C_3F_7CHO$ and $HO_2$ form reaction complexes RC1/RC2, and then pass through transition

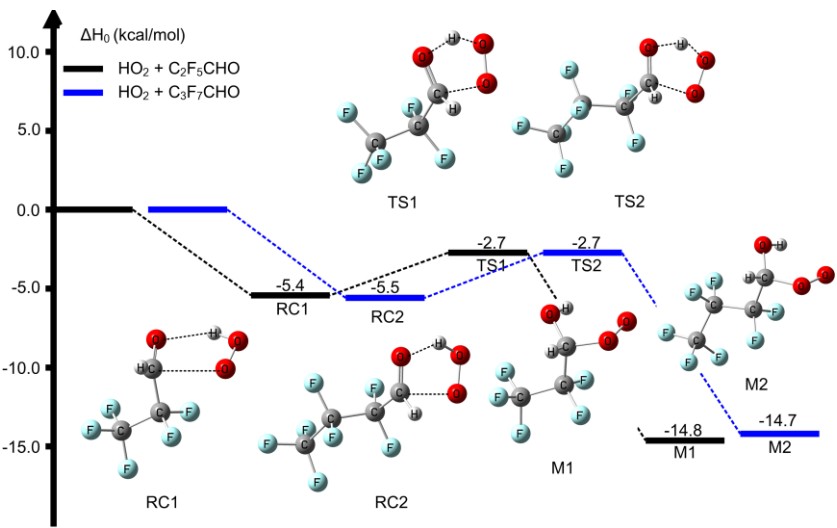

**Figure 1.** Enthalpy profile of the $HO_2$ addition reaction with $C_2F_5CHO$ and $C_3F_7CHO$ as calculated by CCSD(T)-F12a/cc-pVTZ-F12//M06-2X/MG3S level with the scale factor by the standard method at 0 K.



states TS1 and TS2 to form the intermediate products of $C_2F_5CH(OH)OO$ (M1) and $C_3F_7CH(OH)OO$ (M2), respectively. TS1 (–2.7 kcal/mol) denotes the global minimum optimized structures of the stationary points of enthalpy of activation at 0 K; this is 0.3 kcal/mol and 1.6 kcal/mol lower than that of the $CF_3CHO + HO_2$ and $CH_3CHO + HO_2$ reactions, respectively.(Long et al., 2022) This shows that the reaction of perfluoroaldehydes with $HO_2$ may be kinetically feasible. As a comparison, the further energy profile of the $C_3H_7CHO + HO_2$ reaction shows an equal enthalpy of activation of –2.7 kcal/mol for TS2; this is

slightly lower than that of enthalpy of activation –2.4 kcal/mol for the reaction of $CF_3CHO + HO_2$.(Gao et al., 2024b)

It is noteworthy that Figure 1 only depicts the potential energy profile of the reaction featuring the global minimum structure. Nevertheless, the torsion of the C-C bond gives produces multiple conformers for reactants, transition states, and formed intermediate products. Their geometric configurations and energy distributions relative to the global minimum structure are presented in Figure S1. Regarding the reactions of $C_2H_5CHO$ and $C_3H_7CHO$ with $HO_2$, we have observed that as the carbon

chain lengthens, the number of conformers of both reactants and transition states increases, and the energy distribution broadens. For instance, TS1 has three isomers, with an energy distribution spanning from 0 to 1.7 kcal/mol, whereas TS2 has five isomers, and its energy distribution ranges from 0 to 1.9 kcal/mol. In terms of geometric configurations, the low-energy isomers tend to have more linear structures, while the high-energy conformations exhibit more pronounced curling.

NO is a highly reactive gas.(Lee et al., 2024) Human activities, especially agriculture and industrial processes, have led to

significant NO emissions.(Phys et al., 2024; Thomson et al., 2012) Industrial activities contribute to NO levels such as fossil fuel combustion in power plants and chemical manufacturing, along with vehicle emissions. Given its prevalence from human-induced emissions, we further explore the degradation pathways of intermediate products M1 and M2 in the presence of NO. As shown in Figures 2 and S2, M1 and M2 initially react with NO to form the products $C_2F_5CH(O)OH$ and $C_3F_7CH(O)OH$, along with $NO_2$ in the presence of NO. These products then undergo unimolecular reactions to decompose into $C_2F_5$ and $C_3F_7$

radicals and formic acid in Figure 2. Notably, the unimolecular decomposition of $C_2F_5CH(O)OH$ and $C_3F_7CH(O)OH$ represents the rate-determining step of the overall reaction, with corresponding activation enthalpies of 5.6 kcal/mol and 4.7 kcal/mol (0 K), respectively; this indicates that formic acid can be formed via $C_2F_5CHO/C_3F_7CHO + HO_2$ in the presence of high concentration NO in the atmosphere. Additionally, the formed intermediate products (M1 and M2) are a typical class of $RO_2$ radicals. In the low $NO_x$ levels, these $RO_2$ radicals can also participate in bimolecular reactions.(Ding and Long, 2022)

$RO_2$ can react with $HO_2$, resulting in the formation of the stable product ROOH. Moreover, $RO_2$ can react with other $RO_2$ or $R'O_2$ (where R′ denotes a hydrocarbon fragment).(Bottorff et al., 2023) The reaction with $R'O_2$ frequently yields alkoxy radicals, and both of these reactions are capable of producing stable products.(Goldman et al., 2021) Due to the complexity of these bimolecular reactions of the formed $RO_2$ in the reactions R1 and R2, we did not further investigate their reaction mechanisms and kinetics in the present work.





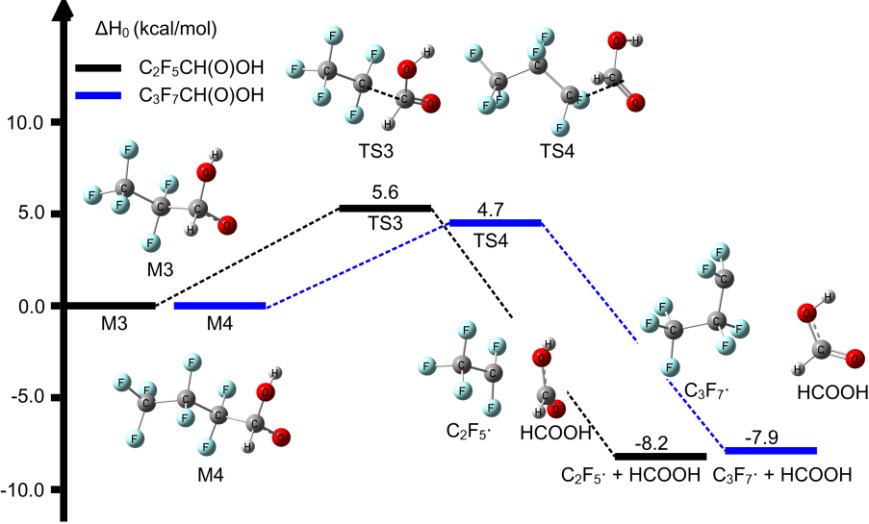


**Figure 2.** Relative enthalpies at 0 K for the decomposition of $C_2F_5CH(O)OH$ (M3) and $C_3F_7CH(O)OH$ (M4) calculated by CCSD(T)-F12a/cc-pVTZ-F12//M06-2X/MG3S.

We further conducted an extended study on the reactions of $C_4F_9CHO$ and $C_5F_{11}CHO$ at the FNO-CCSD(T)-F12//cc-pVDZ-F12//M06-2X/MG3S level, aiming to investigate the effects of increasing carbon chain length on the enthalpy of activation at

0 K. Data from Figure 3 reveal an interesting phenomenon that the activation enthalpy at 0 K remains almost equal C2 ($C_2F_5CHO$) to C5 ($C_5F_{11}CHO$). This finding aligns with the similar trend for the reaction of $C_nH_{2n+1}CHO$ with $HO_2$, suggesting that the impact of carbon chain length growth on the enthalpy of activation at 0 K is quite minor.(Ding and Long, 2022; Gao et al., 2024b) However, the introduction of $CF_3$ leads to a relatively lower enthalpy of activation at 0 K for the $C_nH_{2n+1}CHO$ + $HO_2$ reactions, primarily due to the strong electron-withdrawing ability of fluorine atoms, which can stabilize the transition

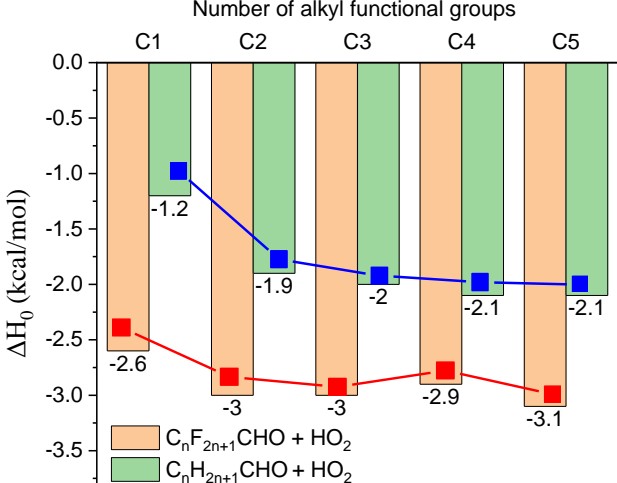

**Figure 3.** The impacts of carbon chain length on the enthalpies of activation at 0 K in the $C_nH_{2n+1}CHO/C_nF_{2n+1}CHO$ + $HO_2$ reactions. The values for $C_nH_{2n+1}CHO$ (n = 1- 5) + $HO_2$ and $C_nF_{2n+1}CHO$ + $HO_2$ are obtained from references (Ding and Long, 2022; Gao et al., 2024a) and calculated by using FNO-CCSD(T)-F12a/cc-pVDZ-F12.




state and lower the enthalpy of activation at 0 K. As the size of perfluoroaldehyde increases, the multi-structure effects caused by torsion of C-C bonds become more pronounced. The relative energy values of reactants and transition states shown in

Figure S1 and S3 (relative to the global minimum energy value, without ZPE correction) indicate that with increasing molecular size, the number of possible isomers increases, leading to a broader energy distribution. This broad energy distribution has significant implications for the thermodynamics and kinetics of the degradation process of perfluoroaldehydes, potentially increasing the diversity and complexity of reaction pathways.

### 3.2. Kinetics of $C_2F_5CHO$/$C_3F_7CHO$ + $HO_2$

The high-pressure limiting rate constants were calculated for the temperature range of 190-350 K, covering a wide atmospheric temperature range. For the reactions $C_2F_5CHO$ + $HO_2$ (R1) and $C_3F_7CHO$ + $HO_2$ (R2), the rate constants incorporating multi-structure anharmonicity corrections are defined as $k_1$ and $k_2$, respectively. According to Zheng (Zheng and Truhlar, 2010), the rate constants at high pressure are fitted using equation (3).

$$k = A \left( \frac{T+T_0}{300} \right)^n \exp \left[ -\frac{E(T+T_0)}{R(T^2 + T_0^2)} \right] \tag{3}$$

Table S4 lists the fitting parameters A, n, E, and $T_0$. Here, T represents temperature in Kelvin, and R is the ideal gas constant (0.0019872 kcal mol$^{-1}$ K$^{-1}$). The temperature-dependent Arrhenius activation energies are determined from the fits using equation (4).

$$E_0 = -R \frac{d \ln k}{d(1/T)} \tag{4}$$

The high-pressure limit rate constants, incorporating multiple-structure anharmonicity torsional corrections, are illustrated in

Figure 4, with more comprehensive data provided in Tables S5-S7. Regarding the reaction involving $C_2F_5CHO$+ $HO_2$, the rate constant $k_1$ exhibits a decrease from $3.35 \times 10^{-12}$ cm$^3$ molecule$^{-1}$ s$^{-1}$ at 200 K to $5.42 \times 10^{-14}$ cm$^3$ molecule$^{-1}$ s$^{-1}$ at 350 K. Similarly, the rate constant $k_2$ for the $C_3F_7CHO$ + $HO_2$ reaction also decreases with increasing temperature.

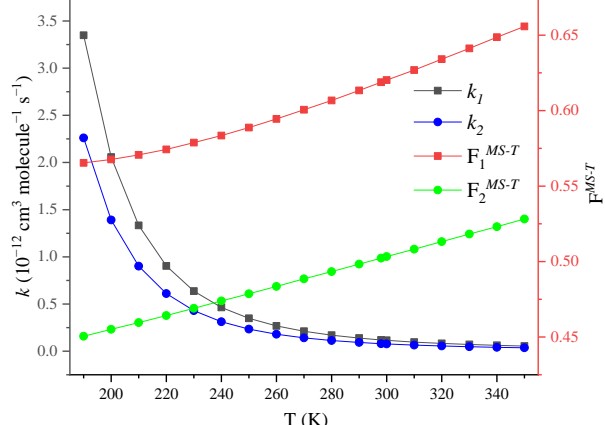

**Figure 4.** The high-pressure limit rate constants of the reactions of $C_2F_5CHO$ and $C_3F_7CHO$ with $HO_2$ at the temperature range of 190–350

K.





In addition, the effects of recrossing and multi-structural anharmonicity are quite limited, approximately ranging between 0.4 and 0.7 times. This results in the rate constants for reactions R1 and R2 being 2-3 times slower than that of $CF_3CHO + HO_2$. For instance, the rate constants of $C_2F_5CHO + HO_2$ and $C_3F_7CHO + HO_2$ are estimated to be $1.19 \times 10^{-13}$ and $7.92 \times 10^{-14}$ cm$^3$ molecule$^{-1}$ s$^{-1}$ at 298 K, respectively, which is slow by compared to $2.48 \times 10^{-13}$ cm$^3$ molecule$^{-1}$ s$^{-1}$ of $CF_3CHO + HO_2$.(Long

et al., 2022) Moreover, the effect of anharmonicity in vibrational-frequency scale factors on high pressure limited rate constants were further discussed. We define "$f$" as the ratio between the rate constant calculated using the reaction-specific vibrational-frequency scale factors and that calculated using the standard vibrational-frequency scale factors. As depicted in Figure 5, the rate constants obtained using the reaction-specific scale factors are 3-7 and 4-10 times fast compared to those calculated using the standard scale factors. Consequently, employing reaction-specific scale factors is crucial for accurate rate calculations.

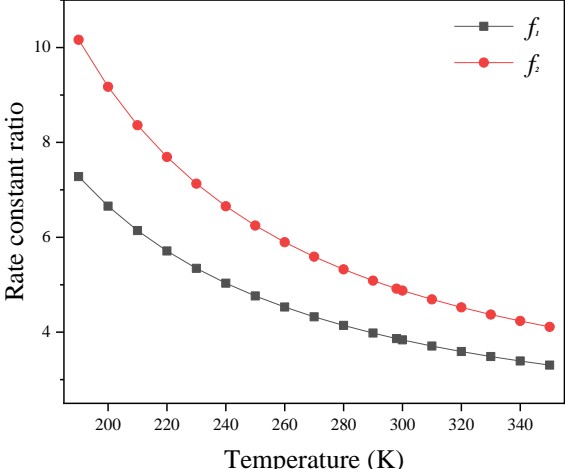


**Figure 5.** The ratio between the rate constant calculated using the reaction-specific vibrational-frequency scale factors and the rate constant calculated using the standard vibrational-frequency scale factors, within the temperature range of 190–350 K. $f_1$ and $f_2$ represent the ratios for the reactions of $C_2F_5CHO + HO_2$ and $C_3F_7CHO + HO_2$, respectively.

The pressure-dependent rate constants of the $HO_2$ reaction with $C_2F_5CHO$ and $C_3F_7CHO$ were further calculated by using SS-

QRRK method. As shown in Figure 6-7 and Table S8-S9, it can be observed that variations of the calculated rate constant with respect to pressure have a minimal impact on the rate constant of $C_2F_5CHO + HO_2$, indicating the absence of significant pressure effects. However, significant pressure effects are observed in the $C_3F_7CHO + HO_2$ reaction, particularly at temperatures above 300 K. To provide a clearer perspective, we define the transition pressure $p_{1/2}$ to quantify the pressure dependence. Specifically, the transition pressure $p_{1/2}$ is the pressure at which the pressure-dependent rate constant reaches half of its high-pressure limit. Figure 8 and Table S10 show that the transition pressure $p_{1/2}$ for the $HO_2 + C_2F_5CHO$ reaction ranging

from $2.6 \times 10^{-5}$ to $7.4 \times 10^{-3}$ bar at 190-350 K, while the transition pressure $p_{1/2}$ ranges from $2.6 \times 10^{-2}$ to 2.3 bar at 190-350 K for the $HO_2 + C_3F_7CHO$ reaction. This indicates that the $HO_2 + C_3F_7CHO$ reaction exhibits a significant pressure dependence, and the increase in carbon chain length has a significantly affect pressure-dependent rate constants.



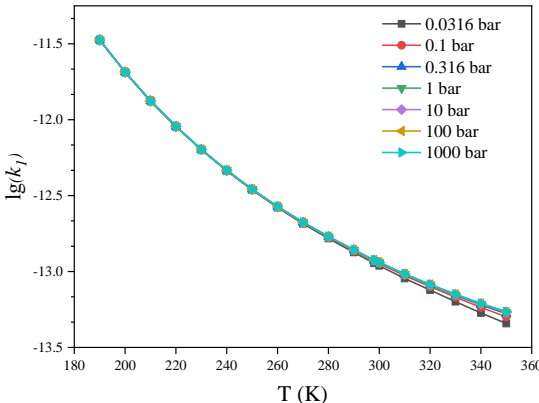

**Figure 6.** Pressure-dependent rate constants of $C_2F_5CHO + HO_2$ as functions of temperature obtained via the SS-QRRK method.

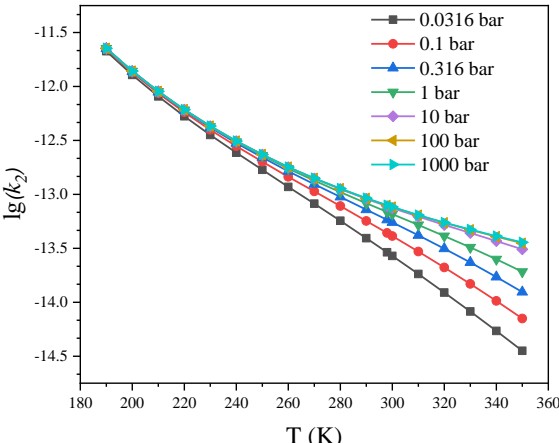

**Figure 7.** Pressure-dependent rate constants of $C_3F_7CHO + HO_2$ as functions of temperature obtained via the SS-QRRK method.

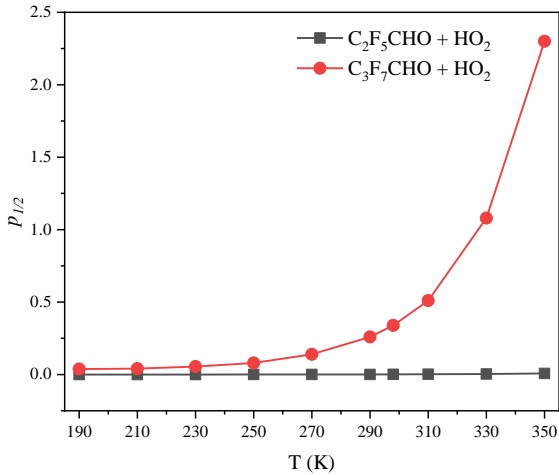

**Figure 8.** Transition pressure $p_{1/2}$ calculated by the SS-QRRK method for the $HO_2 + C_2F_5CHO$ and $HO_2 + C_3F_7CHO$ reactions as functions of temperature.



### 3.3. Atmospheric Implications.

Given that the reaction of OH + $C_2F_5CHO$/$C_3F_7CHO$ was considered to be the main sink for $C_2F_5CHO$/$C_3F_7CHO$, it is necessary to establish the competitive relationship between OH + $C_2F_5CHO$/$C_3F_7CHO$ and $HO_2$+ $C_2F_5CHO$/$C_3F_7CHO$. Therefore, we have defined the ratio of rate between them according to the equations (5) and (6).

$$v_1 = \frac{k_1[C_2F_5CHO][HO_2]}{k_{OH}[C_2F_5CHO][OH]} \tag{5}$$

$$v_2 = \frac{k_1[C_3F_7CHO][HO_2]}{k'_{OH}[C_3F_7CHO][OH]} \tag{6}$$

Here, $k_1$ and $k_2$ are the rate constants of $HO_2$ + $C_2F_5CHO$ and $HO_2$ + $C_3F_7CHO$ calculated by this work, respectively, while $k_{OH}$ and $k'_{OH}$ are the corresponding rate constants of OH+ $C_2F_5CHO$ and OH + $C_3F_7CHO$, from the literature (Solignac et al., 2007; Wang et al., 2007). We calculated the rate ratios using a high OH concentration(Lew et al., 2020) of $5 \times 10^6$ molecules

cm$^{-3}$ and a typical $HO_2$ concentration(Brasseur and Solomon, 2006) of $1.4 \times 10^8$ molecules cm$^{-3}$. The calculations presented in Table 1 reveal that within the temperature range of 240-350 K, the rate ratios for $v_1$ and $v_2$ are in the

**Table 1.** Rate ratios of $HO_2$ + $C_2F_5CHO$ to OH + $C_2F_5CHO$ and $HO_2$ + $C_3F_7CHO$ to OH + $C_3F_7CHO$ within the Temperature Range of 240 to 350 K.

| T(K) | $k_{OH}{}^a$ | $k'_{OH}{}^a$ | $v_1{}^b$ | $v_2{}^b$ |
|---|---|---|---|---|
| 240 | $3.80 \times 10^{-13}$ | $4.30 \times 10^{-13}$ | 34.22 | 20.34 |
| 250 | $4.10 \times 10^{-13}$ | $4.57 \times 10^{-13}$ | 23.80 | 14.31 |
| 260 | $4.40 \times 10^{-13}$ | $4.84 \times 10^{-13}$ | 17.09 | 10.38 |
| 270 | $4.69 \times 10^{-13}$ | $5.10 \times 10^{-13}$ | 12.6 | 7.74 |
| 280 | $4.99 \times 10^{-13}$ | $5.35 \times 10^{-13}$ | 9.55 | 5.90 |
| 290 | $5.28 \times 10^{-13}$ | $5.60 \times 10^{-13}$ | 7.36 | 4.60 |
| 298 | $5.51 \times 10^{-13}$ | $5.80 \times 10^{-13}$ | 6.06 | 3.83 |
| 310 | $5.84 \times 10^{-13}$ | $6.08 \times 10^{-13}$ | 4.62 | 2.96 |
| 320 | $6.12 \times 10^{-13}$ | $6.31 \times 10^{-13}$ | 3.76 | 2.43 |
| 330 | $6.39 \times 10^{-13}$ | $6.54 \times 10^{-13}$ | 3.10 | 2.02 |
| 340 | $6.66 \times 10^{-13}$ | $6.76 \times 10^{-13}$ | 2.59 | 1.70 |
| 350 | $6.92 \times 10^{-13}$ | $6.97 \times 10^{-13}$ | 2.19 | 1.45 |

[a]$k_{OH}$ and $k'_{OH}$ are the rate constants of the OH reactions with $C_2F_5CHO$ and $C_3F_7CHO$, from the literature(Antiñolo et al., 2014; Solignac
et al., 2007) respectively. [b]$v_1$ and $v_2$ denote the rate ratios of $HO_2$ with $C_2F_5CHO$ and $C_3F_7CHO$ to OH with $C_2F_5CHO$ and $C_3F_7CHO$, respectively.

range of 34.22 to 2.19 and 20.34 to 1.45, respectively. These substantial findings indicate that the reactions of $HO_2$ with $C_2F_5CHO$ and $C_3F_7CHO$ can dominate the removal of these compounds, comparing with the corresponding OH reactions, thereby exerting a substantial impact on their elimination processes. We also determined the rate constants governing the



reactions of $C_2F_5CHO$ and $C_3F_7CHO$ with $HO_2$ within the altitude range of 0-50 km, and subsequently evaluated the
atmospheric lifetimes of these compounds. As illustrated in Table 2, the $HO_2$-mediated elimination pathways for $C_2F_5CHO$
and $C_3F_7CHO$ are characterized by relatively rapid reaction rates within the troposphere (< 10 km), resulting in relatively short
atmospheric lifetimes of approximately $5.18 \times 10^4$ - $1.13 \times 10^5$ s (14.4 - 31.3 hours) and $7.78 \times 10^4$ - $1.86 \times 10^5$ s (21.6 - 51.8
hours). These atmospheric lifetimes are considerably shorter than the 20-day atmospheric lifetime estimated for the reactions

involving OH radicals with $C_2F_5CHO$ and $C_3F_7CHO$, and even can somewhat shorter than photolysis (one day)(Solignac et
al., 2007). This highlights the noteworthy impact of $HO_2$-initaited elimination pathways on the atmospheric degradation of
these compounds.

**Table 2.** Hydroperoxyl radical concentration (in molecules cm$^{-3}$), rate constants (in cm$^3$ molecule$^{-1}$ s$^{-1}$), and atmospheric lifetimes (in s) with respect to bimolecular reactions as functions of altitude.

| H (km)[a] | T (K)[a] | P (mbar)[a] | [HO$_2$][b] | $k_1$ (T, $p$)[c] | $k_2$ (T, $p$)[c] | $\tau_1$[d] | $\tau_2$[d] |
|---|---|---|---|---|---|---|---|
| 0 | 290.2 | 1010 | 1.40E+08 | 1.38E-13 | 9.18E-14 | 5.18E+04 | 7.78E+04 |
| 5 | 250.5 | 496 | 4.90E+07 | 3.44E-13 | 2.30E-13 | 5.93E+04 | 8.89E+04 |
| 10 | 215.6 | 243 | 8.30E+06 | 1.07E-12 | 6.47E-13 | 1.13E+05 | 1.86E+05 |
| 15 | 198 | 119 | 2.30E+06 | 2.21E-12 | 2.98E-13 | 1.97E+05 | 1.46E+06 |
| 20 | 208 | 58.2 | 2.90E+06 | 1.38E-12 | 1.22E-13 | 2.50E+05 | 2.83E+06 |
| 25 | 216.1 | 28.5 | 5.70E+06 | 9.38E-13 | 4.32E-14 | 1.87E+05 | 4.06E+06 |
| 30 | 221.5 | 13.9 | 7.50E+06 | 6.52E-13 | 1.29E-14 | 2.05E+05 | 1.04E+07 |
| 35 | 228.1 | 6.83 | 6.90E+06 | 4.04E-13 | 4.13E-15 | 3.59E+05 | 3.50E+07 |
| 40 | 240.5 | 3.34 | 5.90E+06 | 2.29E-13 | 1.73E-15 | 7.41E+05 | 9.82E+07 |
| 45 | 251.9 | 1.64 | 4.90E+06 | 1.27E-13 | 6.56E-16 | 1.60E+06 | 3.11E+08 |
| 50 | 253.7 | 0.801 | 4.00E+06 | 5.87E-14 | 1.80E-16 | 4.26E+06 | 1.39E+09 |

[a]H denotes altitude (atmospheric scale height); T denotes temperature; p denotes pressure. [b]Data are from ref (Brasseur and Solomon, 2006). [c]$k_1$, $k_2$ are the rate constants of the $HO_2$ reactions with $C_2F_5CHO$ and $C_3F_7CHO$, respectively. [d]$\tau_1$, $\tau_2$ are the atmospheric lifetimes of the $HO_2$ reaction with $C_2F_5CHO$ and $C_3F_7CHO$, respectively.

To provide further insight into the importance of these reactions in the atmosphere, we analyse the concentrations of $HO_2$ and
OH by using Geos-Chem. The concentration of modelled $HO_2$ reached a maximum of $4.99 \times 10^8$ molecule cm$^{-3}$ in the Amazon

region and a mean value of $9.93 \times 10^7$ molecule cm$^{-3}$.(Long et al., 2024) Simultaneously, the concentrations of $HO_2$ ($1.7 \times 10^7$
molecule cm$^{-3}$) is consistent with field observations at the British Antarctic Survey's Halley Research Station (1.5 ppt).(Bloss
et al., 2007) We also note that there is a maximum concentration of $8.03 \times 10^6$ molecule cm$^{-3}$ over the Atlantic and Pacific
oceans and an average concentration of $1.06 \times 10^6$ molecule cm$^{-3}$ for OH.(Lelieveld et al., 2016) Based on the rate ratios
calculated above, when the concentration of $HO_2$ is two orders of magnitude higher than that of OH, the reaction with $HO_2$

dominates over the reaction with OH. Therefore, we compare $HO_2$ and OH concentrations in global regions and find at least a



2-order of magnitude difference in concentrations in industrial parks such as Russia, Malaysia, and parts of Africa. (see Figure 9). Especially in the parts of Africa, $HO_2$ is even three orders of magnitude higher than OH. In addition, high concentration

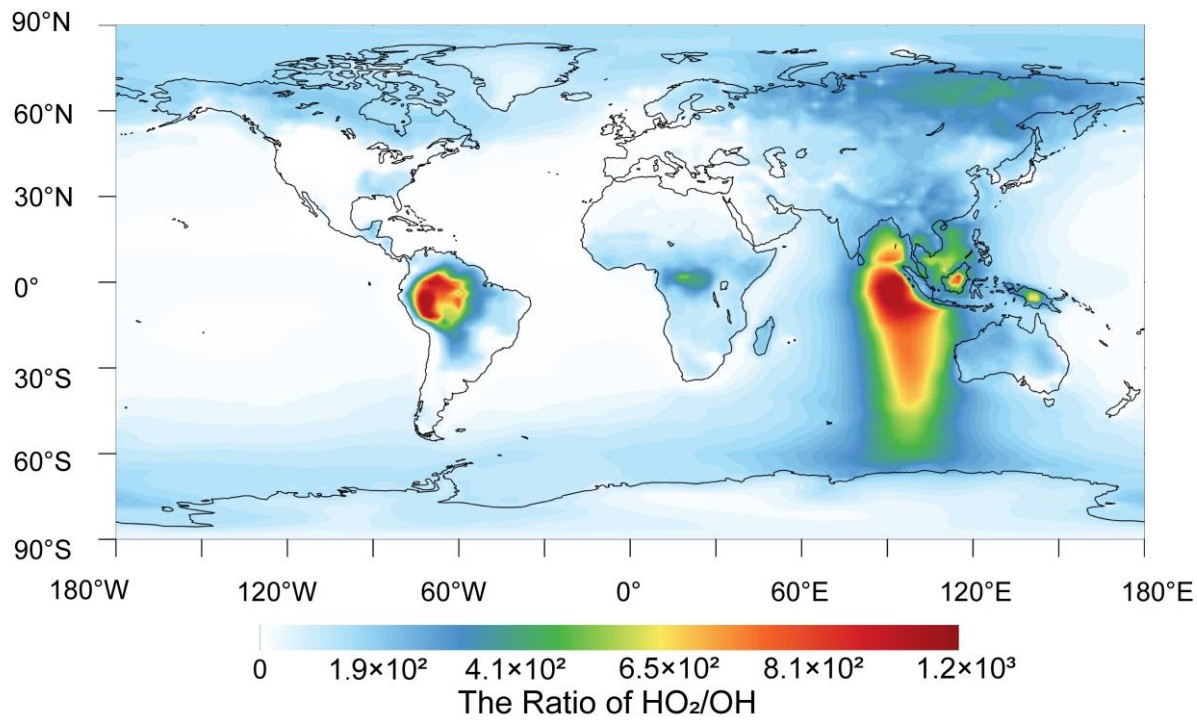

**Figure 9.** The annual average ratio of $HO_2$/OH at night globally.

ratios have been observed along the Indian Ocean margin near Indonesia, which may be due to atmospheric transport. This large concentration ratios between $HO_2$ and OH suggests that $HO_2$ leads to sink of $C_2F_5CHO$ and $C_3H_7CHO$ at night in these particular regions.

Moreover, except for generating formic acid, another intermediate product $C_nF_{2n+1}$ can undergo a single-carbon shortening process as depicted in Figure 10 to ultimately convert into the more stable $COF_2$. Taking the example of $C_2F_5$, the process

starts with $C_2F_5$ reacting with $O_2$ to form $C_2F_5O_2$. Subsequently, $C_2F_5O_2$ reacts with NO to produce $C_2F_5O$, which then undergoes C-C bond cleavage to generate $CF_3$ and $COF_2$. The $CF_3$ further reacts to eventually yield $COF_2$ through a similar reaction pathway.

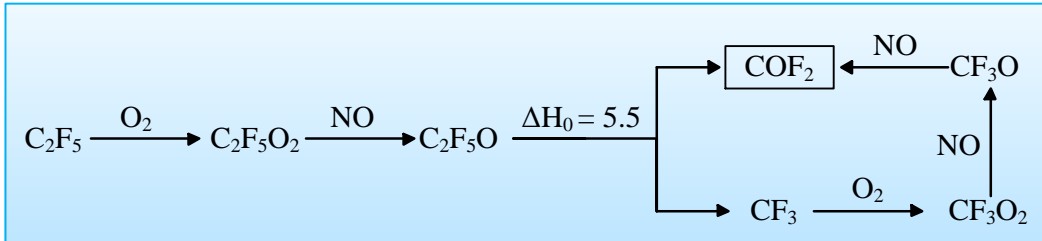

**Figure 10.** Atmospheric degradation mechanism for $C_nF_{2n+1}$ with $C_2F_5$ used as a representative example.



In current studies, we focus on the homogeneous reactions of $HO_2$-initiated linear perfluoroaldehydes. It is worth noting that the concentration of $HO_2$ at interfaces may be higher than in the gas phase, according to research findings.(Angelaki et al., 2024; Li et al., 2023) Given the strong reactivity at the water-air interface and the complex interplay and competition between OH and $HO_2$, the degradation of perfluoroaldehydes initiated by $HO_2$ and their potential contribution to atmospheric acidity may have a more pronounced impact. For example, Xia et al. (2024) recently reported the single-carbon and double-carbon

pathways presented during the degradation of $C_7F_{15}$ on the surface of water droplets. These heterogeneous reactions may also contribute to the removal of perfluoroaldehydes and the degradation process of polyfluoroalkyl substances. Nevertheless, further kinetics and mechanism are still required as a basis and support, so as to enable a more accurate and comprehensive understanding of this complex chemical process and its impact on atmospheric chemistry.

## 4. Summarizing Remarks

In this study, we have delved into the chemical reaction kinetics of linear perfluorinated aldehydes ($C_2F_5CHO$ and $C_3F_7CHO$) with hydroperoxyl radicals in the gas phase using ab initio calculation methods and reaction kinetics theory. We construct a comprehensive reaction potential energy surface and find that the activation enthalpies for the reactions of $C_2F_5CHO$ and $C_3F_7CHO$ with $HO_2$ at 0 K are both -2.7 kcal/mol, which are exactly the same. Our calculated results suggest that the elongation of the carbon chain in linear perfluorinated aldehyde molecules has a negligible effect on the activation enthalpy.

Further kinetic studies reveal that anharmonicity have a significant impact on the reaction rates, while the torsional anharmonicity, recross coefficient, and tunnelling effects contribute relatively little to the rate constants. It is particularly noteworthy that the reaction of $C_3F_7CHO$ with $HO_2$ exhibits a distinct pressure dependence, whereas the reaction of $C_2F_5CHO$ with $HO_2$ does not show such a pressure effect.

Additionally, we compared these reactions with the primary oxidation pathway of linear perfluorinated aldehydes—their

reaction with hydroxyl radicals. We find that there is a big ratio between $HO_2$ and OH concentrations in the Amazon region. The comparative results suggest that the reactions of $HO_2$ with $C_2F_5CHO$ and $C_3F_7CHO$ may dominate their atmospheric chemistry in the Amazon region, thereby affecting the environmental impact of these compounds. Based on our estimates, the atmospheric lifetimes of $C_2F_5CHO$ and $C_3F_7CHO$ are 14.4-31.3 hours and 21.6-51.8 hours, respectively, and under NO conditions this pathway may be a source of HCOOH and $COF_2$ in the troposphere.

Here, we provide new insight into atmospheric degradation of linear perfluorinated aldehydes by $HO_2$ radical. It is noted that further investigations are still required for fully understanding their atmospheric chemical processes. For example, there are another atmospheric oxidant such as Chlorine atom and Criegee intermediates.(Long et al., 2021) We have previously shown that Criegee intermediates can make great contributions to the sinks of formaldehyde and formyl fluoride.(Xia et al., 2023) In addition, the photochemical processes of linear perfluorinated aldehydes are further needed to investigate to

understand photolysis channels and photolysis lifetimes.(Thomson et al., 2024)



**Supplementary Material**

Separate document

**Author contributions.**

BL designed the project; ZGD performed the quantum chemical calculations; CLX performed the model calculations; ZGD, CLX, and BL analysed the data; ZGD wrote the manuscript draft. ZGD, CLX, and BL reviewed and edited the manuscript.

**Competing interests**

The authors declare that they have no conflict of interest.

**Financial support**

This work was supported in part by the National Natural Science Foundation of China (42120104007 and 41775125) and by Guizhou Provincial Science and Technology Projects, China (ZK [2022]194, CXTD [2022]001, and GCC [2023]026).

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
