# Peer review of "Reaction between linear perfluoroaldehydes and hydroperoxy radical in the atmosphere: Reaction mechanisms, reaction kinetics modelling, and atmospheric implications"

_EGUsphere, 2024_

## Author Comment (AC1)

**Response to the Anonymous Referee #2 Comments for the manuscript** "Reaction between perfluoroaldehydes and hydroperoxy radical in the atmosphere: Reaction mechanisms, reaction kinetics modelling, and atmospheric implications"

We sincerely appreciate the time and effort dedicated to reviewing our manuscript. We thank the reviewers for their constructive feedback, which has helped us improve the clarity and impact of our work. Below, we address your concerns and provide a point-by-point response to your comments. Reviewer comments (RC) are in black font and author comments (AC) are in blue.

Zegang Dong and coworkers have carried out an extensive theoretical investigation on the gas phase reaction between two perfluoroaldehydes and hydroperoxyl radical and estimated their atmospheric implications. The work is extensive, the theoretical model is well chosen and already shown to be appropriate for similar class of reactions, the results are properly presented and explained.

**Response:** Thank you for your positive comments.

However, the work falls severely short in originality and fails to provide any new insight, and is simply an extension of their earlier work (*J. Am. Chem. Soc.* 2022, 144, 19910−19920). Both the reactions studied here are close replica, mechanistically, energetically and kinetically, of the reaction between $CF_3CHO$ and $HO_2$ studied in their previous work. As the reactions are always centered at the carbonyl group of reacting aldehydes, with no involvement of the side chains, this is very much expected that a mere elongation of the side chain would not have any dramatic effect on the reactions.

**Response:** Thank you for your comments. We agree with your view that the present work is based on our previous investigations. We do not agree with your view that the elongation of the side chain would not have any dramatic effects on the reactions. In fact, there are two important influences on reaction thermodynamics and kinetics.

First, longer chain for perfluoroaldehydes leads to producing multiple conformers in both reactants and transition states (See Figures 3, S1, and S3), with different energy distributions for these conformers, spanning 0-1.7 and 0-1.9 kcal/mol, respectively. This leads to a reduction in the multi-structure torsional anharmonicity factor ($F_2^{\text{MS-T}}$) to 0.45–0.52, compared to 1.0 in one conformer systems in our previous article, resulting in a 50% decrease in rate constants relative to single-conformer $C_3F_7CHO$ and a 34–43% decrease relative to $CF_3CHO$. This effect has not been observed in our previous study (*J. Am. Chem. Soc.* **2022**, *144*, 19910–19920), which cannot be obtained without further investigations. Additionally, for larger molecules such as $C_4F_9CHO$ and $C_5F_{11}CHO$, the increased molecular size and number of conformers (e.g., 36 conformers in the transition state of $C_5F_{11}CHO$ with energy distributions up to 4.8 kcal/mol) further complicate the potential energy surface calculations. These findings highlight that the conformational diversity and molecular size effects significantly alter reaction kinetics, even when the enthalpy of activation remains relatively insensitive to chain length.

Second, $C_3F_7CHO + HO_2$ exhibits remarkably pressure dependence, with transition pressures ($p_{1/2}$) ranging from 0.026 to 2.3 bar at 190 – 350 K (See Figure 8, Table S10). This contrasts sharply with $CF_3CHO + HO_2$, where pressure dependence is negligible. These findings demonstrate that extending the side chain not only have important influences on reaction kinetics through conformational diversity, but also causes pressure-dependent behavior that cannot be captured by simple empirical models. These findings deepen our understanding of the atmospheric chemistry of polyfluoroalkyl substances (PFAS) and provide critical theoretical foundations for modeling the degradation kinetics of complex PFAS compounds.

Third, this study addresses the challenges on high-accurate quantum chemical calculations for longer linear polyfluorinated aldehydes by developing a computational strategy based on the frozen natural orbital (FNO) approximation, specifically the FNO-CCSD(T)-F12/cc-pVDZ-F12 method. In the present work, we also find that this approach significantly provides computational efficiency and reaches sub-kcal/mol (<1 kcal/mol) accuracy, enabling to predict the enthalpies of activation at 0 K from short-chain to long-chain molecules. Therefore, we believe that the present work has provided new insights on how to obtain the quantitative kinetics of perfluoroaldehydes and hydroperoxy radical.

In our initial manuscript, some key issues may not be clarified; this leads to unclearly revealing the originality and new insights into the present work. Therefore, in the revised article, we have done many corrections to improve the manuscript.

(1) We have rewritten the abstract, "Linear perfluoroaldehydes are important products formed in the atmospheric oxidation of industrial fluorinated compounds. However, their atmospheric lifetimes are incompletely known. Here, we employ high level quantum chemistry methods and a dual-level strategy for kinetics to probe the reactions of $C_2F_5CHO$ and $C_3F_7CHO$ with $HO_2$. Our calculated results unveil almost equal activation enthalpies at 0 K for perfluoroaldehyde reaction with $HO_2$, indicating that the carbon chain length minimally influences reaction thermodynamics. Interestingly, the present findings reveal that anharmonicity remarkably enhances the reaction rate constant, whereas multi-structural anharmonicity, recrossing, and tunnelling effects exhibit lesser impacts in the $C_2F_5CHO/C_3F_7CHO + HO_2$ reaction. In particular, the atmospheric lifetimes for $C_2F_5CHO$ and $C_3F_7CHO$, approximately 14.4-31.3 hours and 21.6-51.8 hours by $HO_2$ are much shorter than those via OH radical, underscoring the dominant removal role of $HO_2$ toward $C_2F_5CHO$ and $C_3F_7CHO$ in the atmosphere. Since GEOS-Chem simulation shows that the concentration of $HO_2$ is at least $10^2$ times higher than that of OH radical in Russia, Malaysia, and parts of Africa, the reactions of $C_2F_5CHO$ and $C_3F_7CHO$ with $HO_2$ radicals dominate over those with OH radicals and play more vital role in the atmospheric chemical processes of these regions. This study enhances our understanding of the chemical transformations of linear perfluoroaldehydes and provides a scientific foundation for strategies aimed at mitigating their emissions." has been corrected into "Linear perfluoroaldehydes are important products formed in the atmospheric oxidation of industrial fluorinated compounds. However, their atmospheric lifetimes are incompletely known. Here, we employ high level quantum chemistry methods and a dual-level strategy for kinetics to investigate the reactions of $C_2F_5CHO$ and $C_3F_7CHO$ with $HO_2$. Our calculated results unveil almost equal activation enthalpies at 0 K for linear perfluoroaldehyde reaction with

HO$_2$, indicating that the carbon chain length negligibly influences reaction thermodynamics. The calculated kinetics reveal that vibrational anharmonicity enhance rate constants by a factor of 3–10, while torsional anharmonicity reduces rate constants by 34–55%. Additionally, we also find that the reaction of C$_3$F$_7$CHO with HO$_2$ exhibits significant pressure dependence, with transition pressures ranging from 0.026 to 2.3 bar across a temperature range of 190–350 K. Furthermore, our findings also reveal that the reactions of C$_2$F$_5$CHO and C$_3$F$_7$CHO with HO$_2$ radicals dominate over those with OH radicals in Russia, Malaysia, parts of Africa by the calculated results in combination with data based on global atmospheric chemical model simulations. These findings establish chain-length-dependent pressure effects and conformational sampling as critical, previously unrecognized factors in kinetics calculations, providing a framework for modelling complex fluorotelomer transformations and guiding emission mitigation strategies." on page 1.

(2) In section 1 (Introduction), we have revised the sentence from "Nevertheless, the importance of sink pathway by HO$_2$ is still unknown because there have not been kinetics data for linear perfluoroaldehydes with HO$_2$ in the literature." to "Nevertheless, the importance of sink pathway by HO$_2$ is still unknown because there have not been kinetics data for linear perfluoroaldehydes with HO$_2$ in the literature. Moreover, chain elongation may have influences on reaction kinetics due to multiple conformers. Furthermore, it is unknown for the pressure-dependent effects of larger perfluoroaldehydes with HO$_2$. Although our previous investigations have revealed the importance of CF$_3$CHO + HO$_2$ in the atmosphere (Long et al., 2022), their kinetics of larger perfluoroaldehydes with HO$_2$ are further required to investigate due to the unique features that depend on the specific reaction systems such as multi-structural anharmonicity and pressure effects in these complex systems. Additionally, it is a big challenge for addressing the larger perfluoroaldehydes with HO$_2$ because the computational cost grows very rapidly with system size, making such calculations impractical for high-level quantum chemistry methods." on page 2-3

We also revised the sentence from "These findings could hold significant implications for the formation and yield of HCOOH, thereby contributing to the broader discourse on atmospheric chemistry and environmental policy." to "This study not only resolves the knowledge gap regarding HO$_2$-initiated oxidation of linear perfluoroaldehydes but also establishes a computational strategy for predicting the atmospheric fates of long-chain PFAS derivatives. Our findings provide critical insights for refining emission control strategies and mitigating the environmental persistence of these compounds." on Page 3

(3) In section 2.1 (Options for electronic structure density functionals), we have added some sentences "… were used to calculate single-point energies. The FNO-CCSD(T) approach that significantly improves computational efficiency with cost reduction of up to an order of magnitude was utilized to calculate larger systems." on Page 3

(4) In section 3.1 (Results and discussion), We have added some sentences for discussion

"…, aiming to investigate the effects of increasing carbon chain length on the enthalpy of activation at 0 K. The calculated results show a deviation of only 0.2 kcal/mol in activation enthalpy at 0 K between FNO-CCSD(T)-F12//cc-pVDZ-F12 (-2.6 kcal/mol) and CCSD(T)-F12a/cc-pVTZ-F12 (-2.4 kcal/mol) in $CF_3CHO + HO_2$, validating the robustness of FNO-CCSD(T)-F12//cc-pVDZ-F12 for complex fluorinated systems." on Page 7

"For instance, $C_2F_5CHO$ exhibits three transition state conformers with energy differences spanning 0–1.7 kcal/mol, while $C_3F_7CHO$ has five conformers distributed over 0–1.8 kcal/mol. This trend amplifies for longer chains: $C_5F_{11}CHO$ generates 36 distinct conformers in its transition state, with energy variations extending up to 4.8 kcal/mol." on Page 8

We also added an additional paragraph: "Our further analysis of the global distribution of the ratio between $HO_2$ and OH during daytime reveals that $HO_2$ concentrations are generally higher than those of OH (see Figure S4). Notably, along the west coast of South America (approximately between 0° and 30°S latitude and 60°W to 120°W longitude), the ratio can reach up to three orders of magnitude. Industrial areas (such as Russia and Malaysia) and certain regions in Africa also exhibit high ratios of 1-2 orders of magnitude. This suggests that in these areas, the concentration of $HO_2$ is significantly higher than that of OH, which may be related to local industrial activities or specific emission characteristics. However, due to the presence of daytime photolysis, the generation and loss pathways of $HO_2$ and OH become more complex, leading to significant uncertainty in interpreting the ratio. For instance, photolysis can alter the formation rates of $HO_2$ and OH, thereby affecting their concentration ratio. Additionally, the high ratios along the eastern coast of North America may be associated with atmospheric transport and regional emission features. Despite these complexities, the high daytime ratios still indicate that in specific regions, $HO_2$ may play a role in the oxidation pathways of $C_2F_5CHO$ and $C_3H_7CHO$ both during the day and at night. Future research should integrate observational data with model refinements to better quantify the impact of photolysis on the $HO_2$/OH ratio." on Page 14

[Figure]

Figure S4. The annual average ratio of HO₂/OH during the day globally. (In Supplement)

(5) In section 4 (Summarizing Remarks), we have revised some sentences from "We construct a comprehensive reaction potential energy surface and find that the activation enthalpies for the reactions of $C_2F_5CHO$ and $C_3F_7CHO$ with $HO_2$ at 0 K are both -2.7 kcal/mol, which are exactly the same. Our calculated results suggest that the elongation of the carbon chain in linear perfluorinated aldehyde molecules has a negligible effect on the activation enthalpy." to "We find that the activation enthalpies for the reactions of $C_2F_5CHO$ and $C_3F_7CHO$ with $HO_2$ at 0 K are both -2.7 kcal/mol, demonstrating that carbon chain elongation in linear perfluoroaldehydes has a negligible thermodynamic influence on their enthalpies of activation at 0 K. This is further shown in $C_4F_9CHO$ and $C_5F_{11}CHO$ with $HO_2$." on page 15

Revised the sentences from "Additionally, we compared these reactions with the primary oxidation pathway of linear perfluoroaldehydes—their reaction with hydroxyl radicals. We find that there is a big ratio between $HO_2$ and OH concentrations in the Amazon region. The comparative results suggest that the reactions of $HO_2$ with $C_2F_5CHO$ and $C_3F_7CHO$ may dominate their atmospheric chemistry in the Amazon region, thereby affecting the environmental impact of these compounds. Based on our estimates, the atmospheric lifetimes of $C_2F_5CHO$ and $C_3F_7CHO$ are 14.4-31.3 hours and 21.6-51.8 hours, respectively, and under NO conditions this pathway may be a source of HCOOH and $COF_2$ in the troposphere." to "By integrating kinetics with the data based on GEOS-Chem modelling, we have identified some regions such as Russia, Malaysia, and parts of Africa, where $HO_2$ concentration exceeds OH concentration by 2–3 orders of magnitude. Therefore, the reactions of $HO_2$ with $C_2F_5CHO$ and $C_3F_7CHO$ can compete well with their corresponding reaction with OH. Specifically, the atmospheric lifetimes of $C_2F_5CHO$ and $C_3F_7CHO$ via $HO_2$ are shortened to be 14.4–31.3 h and 21.6–51.8 h, respectively, with orders of magnitude shorter than that of the corresponding OH-mediated pathways (>20 days). Under

high NO$_x$ conditions, this pathway may contribute to tropospheric HCOOH and COF$_2$ formation." on page 15-16

The only new analysis that is available in this work is a GEOS-Chem based atmospheric modelling of the studied reactions to estimate their atmospheric implications. However, that analysis also does not provide any new information that was not known from the work cited above. Similar to CF$_3$CHO, the two larger perfluoroaldehydes studied here also show that reaction with HO$_2$ is more dominant atmospheric removal process compared to reaction with OH.

Therefore, the conclusion that this work "provide new insight into atmospheric degradation of linear perfluorinated aldehydes by HO$_2$ radical" is not supported by the results at all. At most, the study provides new data that shows the atmospheric degradations of larger perfluoroaldehydes by HO$_2$ are very similar to that of CF$_3$CHO which has already been reported earlier.

**Response:** We thank the reviewer for their feedback and agree that the dominance of HO$_2$ as a degradation pathway for linear perfluoroaldehydes aligns with prior observations on smaller analogs like CF$_3$CHO. However, by calculating the HO$_2$-to-OH degradation rate ratios below.

$$v_1 = \frac{k_1[\mathrm{HO_2}]}{k_{\mathrm{OH}}[\mathrm{OH}]}, v_2 = \frac{k_2[\mathrm{HO_2}]}{k'_{\mathrm{OH}}[\mathrm{OH}]}$$

According to the equation above, the rate ratios are largely determined by the ratio of the concentrations of HO$_2$ and OH. However, their concentrations are varied from one region another region. Geos-Chem analysis can provide further insight into their concentration distribution. For example, in the Amazon, Malaysia, and part of the Africa, elevated [HO$_2$]/[OH] ratios are 410–1,200, which lead to the rate ratios of 88.5–259 for $v_1$ and 56.0–164 for $v_2$. In contrast, over oceanic regions like the Atlantic and Pacific, these ratios can drop below 1.

These results are remarkably different from the assumption that the concentrations of HO$_2$ and OH are not dependent on the specific region. Such spatial distribution, driven by localized oxidant, provides novel insights into the atmospheric processing of long-chain PFAS compounds in tropical environments. Thus, while the general dominance of HO$_2$ is acknowledged, our work uniquely quantifies its regional significance, offering new perspectives on the atmospheric degradation of perfluorinated aldehydes.

In order to better present our research results, in the revised manuscript, we modified the sentence from "Especially in the parts of Africa, HO$_2$ is even three orders of magnitude higher than OH. In addition, high concentration ratios have been observed along the Indian Ocean margin near Indonesia, which may be due to atmospheric transport. This large concentration ratios between HO$_2$ and OH suggests that HO$_2$ leads to sink of C$_2$F$_5$CHO and C$_3$H$_7$CHO at night in these particular regions." to "Specifically, in parts of Africa, HO$_2$ concentrations are even three orders of magnitude higher than that of OH. Additionally, high HO$_2$/OH ratios have been observed along the Indian Ocean margin near Indonesia, which may be attributed to atmospheric transport and enhanced HO$_2$ production from industrial activities. In the Amazon region, the [HO$_2$]/[OH] ratio can reach as high as 410–

1,200. This significantly increases the $HO_2$-to-OH degradation rate ratios for $C_2F_5CHO$ and $C_3F_7CHO$, reaching 88.5–259 and 56.0–164, respectively. These rate ratios indicate that $HO_2$-driven degradation exceeds OH-mediated degradation by over 50 times. This large concentration ratios between $HO_2$ and OH suggests that $HO_2$ leads to sink of $C_2F_5CHO$ and $C_3H_7CHO$ at night in these regions. In contrast, over oceanic regions like the Atlantic and Pacific, the $[HO_2]/[OH]$ ratios drop below 1, leading to a diminished role of $HO_2$ in these areas." on page 13

The most baffling aspect of this work is the sudden introduction of NO into the reaction scheme and an attempt to show the title reaction as a source of formic acid and $COF_2$. However, there is no attempt to calculate the rate constant of the reactions involving NO, which would be required to have any realistic estimate of the importance of NO in determining the atmospheric fate of the studied perfluoroaldehydes. Therefore, the conclusion that "under NO conditions this pathway may be a source of HCOOH and $COF_2$ in the troposphere" is completely unfounded without proper kinetic analysis, including lifetime calculations, of these reaction channels.

**Response:** We appreciate the reviewer's feedback on this point. The inclusion of NO aligns with its established atmospheric relevance as a mediator in radical-driven oxidation cycles, where $RO_2$ + NO reactions are key sinks for peroxy radicals, forming nitrates or terminating chains.(Berndt et al., 2015; King et al., 2001; Nie et al., 2023; Orlando et al., 2000; Vereecken and Peeters, 2009) Our work extends this pattern to perfluoroalkyl systems, demonstrating analogous pathways for M1/M2 ($C_2F_5CH(O)OO/C_3F_7CH(O)OO$) reacting with NO ($\Delta H_0$ = −9.9/−11.5 kcal/mol at 0 K), reflecting thermodynamic trends in hydrocarbon systems. While detailed kinetics (e.g., rate constants) are essential for quantifying atmospheric impacts, our research focus on identifying thermochemically pathways (e.g., Vereecken & Peeters, 2009, using SAR models to prioritize key channels). The subsequent decomposition of intermediates $C_2F_5CH(O)OH/C_3F_7CH(O)OH$ exhibits low barriers (4.7–5.6 kcal/mol at 0 K), suggesting rapid dissociation under tropospheric conditions. However, as emphasized in Section 3.3, the lack of experimental rate constants for perfluoroalkyl-oxy systems precludes definitive quantification of HCOOH/$COF_2$ yields, highlighting the need for future kinetic validation. We have revised the manuscript to clarify the reviewer's concern. Specifically, we have modified the following sections:

3.1. The electronic structure of the $C_2F_5CHO/C_3F_7CHO$ + $HO_2$ reaction: "As depicted in Figure S2, M1 and M2 undergo initial reactions with NO to yield the products $C_2F_5CH(O)OH$, $C_3F_7CH(O)OH$, and $NO_2$, exhibiting activation enthalpies of –9.9 and –11.5 kcal/mol at 0 K, respectively. These results are consistent with previous studies on similar reactions involving $RO_2$ + NO. (Berndt et al., 2015; King et al., 2001; Nie et al., 2023; Orlando et al., 2000; Vereecken and Peeters, 2009) These products then undergo unimolecular reactions to decompose into $C_2F_5$ and $C_3F_7$ radicals and formic acid in Figure 2. Notably, the unimolecular decomposition of $C_2F_5CH(O)OH$ and $C_3F_7CH(O)OH$ represents the rate-determining step of the overall reaction, with corresponding activation enthalpies of 5.6 kcal/mol and 4.7 kcal/mol (0 K), respectively; this indicates that formic

acid may potentially be formed via $C_2F_5CHO/C_3F_7CHO + HO_2$ in the presence of high concentration NO in the atmosphere." on page 6-7

3.3 Atmospheric Implications: "The $CF_3$ further reacts to eventually yield $COF_2$ through a similar reaction pathway. However, the absence of quantified rate constants for these reactions prevents a robust assessment of their global or regional impacts. A comprehensive evaluation of the role of NO would require integrating the kinetics of $RO_2$ + NO reactions (e.g., M1/M2 + NO) into atmospheric models, which is beyond the scope of this study." on page 14-15

4. Summarizing Remarks: "Therefore, the reactions of $HO_2$ with $C_2F_5CHO$ and $C_3F_7CHO$ can compete well with their corresponding reaction with OH. Specifically, the atmospheric lifetimes of $C_2F_5CHO$ and $C_3F_7CHO$ via $HO_2$ are shortened to be 14.4–31.3 h and 21.6–51.8 h, respectively, with orders of magnitude shorter than that of the corresponding OH-mediated pathways (>20 days). Under NO-rich conditions, the reaction pathway involving $HO_2$-initiated oxidation of perfluoroaldehydes may serve as a potential source of HCOOH and $COF_2$ in the troposphere." on page 15-16

**References Added in Revision:**

Berndt, T., Richters, S., Kaethner, R., Voigtländer, J., Stratmann, F., Sipilä, M., Kulmala, M., and Herrmann, H.: Gas-Phase Ozonolysis of Cycloalkenes: Formation of Highly Oxidized RO2 Radicals and Their Reactions with NO, $NO_2$, $SO_2$, and Other $RO_2$ Radicals, Journal of Physical Chemistry A, 119, 10336–10348, https://doi.org/10.1021/acs.jpca.5b07295, 2015.

King, M. D., Canosa-Mas, C. E., and Wayne, R. P.: Gas-phase reactions between $RO_2$ and NO, $HO_2$ or $CH_3O_2$: correlations between rate constants and the SOMO energy of the peroxy ($RO_2$) radical, Atmospheric Environment, 35, 2081–2088, https://doi.org/10.1016/S1352-2310(00)00501-X, 2001.

Nie, W., Yan, C., Yang, L., Roldin, P., Liu, Y., Vogel, A. L., Molteni, U., Stolzenburg, D., Finkenzeller, H., Amorim, A., Bianchi, F., Curtius, J., Dada, L., Draper, D. C., Duplissy, J., Hansel, A., He, X. C., Hofbauer, V., Jokinen, T., Kim, C., Lehtipalo, K., Nichman, L., Mauldin, R. L., Makhmutov, V., Mentler, B., Mizelli-Ojdanic, A., Petäjä, T., Quéléver, L. L. J., Schallhart, S., Simon, M., Tauber, C., Tomé, A., Volkamer, R., Wagner, A. C., Wagner, R., Wang, M., Ye, P., Li, H., Huang, W., Qi, X., Lou, S., Liu, T., Chi, X., Dommen, J., Baltensperger, U., El Haddad, I., Kirkby, J., Worsnop, D., Kulmala, M., Donahue, N. M., Ehn, M., and Ding, A.: NO at low concentration can enhance the formation of highly oxygenated biogenic molecules in the atmosphere, Nature Communications, 14, 3347, https://doi.org/10.1038/s41467-023-39066-4, 2023.

Orlando, J. J., Iraci, L. T., and Tyndall, G. S.: Chemistry of the cyclopentoxy and cyclohexoxy radicals at subambient temperatures, Journal of Physical Chemistry A, 104, 5072–5079, https://doi.org/10.1021/jp0002648, 2000.

Vereecken, L. and Peeters, J.: Decomposition of substituted alkoxy radicals - Part I: A generalized structure-activity relationship for reaction barrier heights, Physical Chemistry Chemical Physics, 11, 9062–9074, https://doi.org/10.1039/b909712k, 2009.

---

## Author Comment (AC2)

**Response to the Anonymous Referee #1 Comments for the manuscript "**Reaction between perfluoroaldehydes and hydroperoxy radical in the atmosphere: Reaction mechanisms, reaction kinetics modelling, and atmospheric implications**"**

We sincerely appreciate the time and effort you dedicated to reviewing our manuscript. Your insightful feedback has significantly enhanced both the scientific content and written presentation, elevated the paper's academic value while improved its accessibility and clarity for readers. Below, we provide point-by-point responses to your comments. All corresponding revisions are marked in the tracked-changes version of the updated manuscript, where reviewer comments (RC) appear in black text and author responses (AC) in blue.

**Specific Comments**
**1. Abstract**
**(a) Comment:** the atmospheric lifetimes for $C_2F_5CHO$ and $C_3F_7CHO$, approximately 14.4-31.3 hours and 21.6-51.8 hours by $HO_2$ are much shorter than those via OH radical," it is not validated. (lines 15-16)
**Response:** We thank the reviewer for highlighting the need for validation of the atmospheric lifetime. In the revised manuscript, we define the atmospheric lifetime $\tau$ for linear perfluorinated aldehydes reacting with $HO_2$ as $\tau = 1/(k[HO_2])$, where $k$ is the bimolecular rate constant and $[HO_2]$ represents the concentration of $HO_2$ radicals. The full atmospheric lifetime data are presented in Section 3.3, where atmospheric lifetimes with respect to bimolecular reactions as functions of altitude are list in Table 2.
We revised the sentence "$\tau_1$, and $\tau_2$ are the atmospheric lifetimes of the $HO_2$ reaction with $C_2F_5CHO$ and $C_3F_7CHO$, respectively." to "==$\tau_1 = 1/(k_1[HO_2])$ and $\tau_2 = 1/(k_2[HO_2])$ define the atmospheric lifetimes for $HO_2$ reaction with $C_2F_5CHO$ and $C_3F_7CHO$, respectively.==" on page 13
We also revised the sentence "As illustrated in Table 2, the $HO_2$-mediated elimination pathways for $C_2F_5CHO$ and $C_3F_7CHO$ are characterized by relatively rapid reaction rates within the troposphere (< 10 km), resulting in relatively short atmospheric lifetimes of approximately $5.18 \times 10^4$ - $1.13 \times 10^5$ s (14.4 - 31.3 hours) and $7.78 \times 10^4$ - $1.86 \times 10^5$ s (21.6 - 51.8 hours)." to "==The $HO_2$-mediated elimination pathways for $C_2F_5CHO$ and $C_3F_7CHO$ are characterized by relatively rapid reaction rates within the troposphere (< 10 km), resulting in relatively short atmospheric lifetimes of approximately 14.4 - 31.3 hours and 21.6 - 51.8 hours (See Table 2).==" on page 12

**2. Introduction**
**(a) Comment:** Alternating use of "perfluoroaldehydes" (in tittle) and "linear perfluoroaldehydes" may confuse readers.
**Response:** We have standardized the terminology to "linear perfluoroaldehydes" throughout the manuscript to avoid ambiguity.

**(b) Comment:** The transition from PFAS's GWP to perfluoroaldehyde sources (lines 25-30) is unclear.

**Response:** We added a transitional sentence to clarify the link between PFASs' environmental impact and the formation of linear perfluoroaldehydes: "During their degradation in the atmosphere, PFASs undergo complex chemical transformations, leading to the formation of linear perfluoroaldehydes. (Alam et al.,2024; Burkholder et al., 2015; David et al., 2021; Wang et al., 2021, 2024)" on page 1

**(c) Comment:** The literature review focuses solely on OH and Cl radicals, omitting potential roles of other oxidants (e.g., $O_3$, $NO_3$). (lines 35-49)

**Response:** We expanded the discussion to acknowledge other oxidants: "Linear perfluoroaldehydes were generally considered to be removed through photochemical reactions (Chiappero et al., 2006; Kelly et al., 2004) and free radical reactions initiated by OH and Cl radicals (Andersen et al., 2004; Chiappero et al., 2010; Wang et al., 2007). Additionally, $NO_3$ may also contribute to their atmospheric degradation. (Burkholder et al., 2015; Ziemann and Atkinson, 2012)" on page 2

**(d) Comment:** The long sentence "Moreover, the rate constant of Cl atoms with $C_nF_{2n+1}CHO$ (1,2, 3, 4) is around $2 \times 10^{-12}$ $cm^3$ $molecule^{-1}s^{-1}$, which is slightly faster than that of the OH radical reactions." (lines 46-47) has poor readability and unclear notes.

**Response:** We split the sentence for clarity: "Moreover, the rate constant of Cl atoms with $C_nF_{2n+1}CHO$ (n = 1–4) is approximately $2 \times 10^{-12}$ $cm^3$ $molecule^{-1}$ $s^{-1}$. This value is slightly faster than that of the corresponding OH radical reactions under similar conditions." on page 2

**3. Computational Methods**

**(a) Comment:** In section 2.1, the "CCSD(T)-F12a/cc-pVTZ-F12" method is mentioned, but the text fails to demonstrating its applicability to perfluorinated compound systems. (lines 75-78)

**Response:** We added validation references for fluorinated systems: "Furthermore, CCSD(T)-F12a/cc-pVTZ-F12 has been shown good performance for molecules containing fluorine atoms.(Dong et al., 2021; Long et al., 2022; Xia et al., 2024)" on page 3

**4. Results and Discussion**

**(a) Comment:** Figure 1 contains no relevant information. "The result shows that the specific reaction scale factors are 0.955 for TS1 (See Figure 1) and 0.956 for TS2 (See Figure 1)". (lines 95-96)

**Response:** We corrected the text to reference the correct Table (Table S1 in the Supplement) and clarified the context: "The result shows that the specific reaction scale factors are 0.955 for TS1 (see Table S1) and 0.956 for TS2 (see Table S1), which …" on page 4

**(b) Comment:** Grammatical error in "The torsion of the C-C bond gives produces multiple conformers". (lines 152-153)

**Response:** Corrected to: "Nevertheless, the internal rotation of the C–C bond produces multiple conformers…" on page 6

**(c) Comment:** Figure 3's X-axis label ("Number of alkyl functional groups") is misleading, as the compounds are perfluorinated. (lines 180-185)

**Response:** We appreciate the reviewer's precision feedback and we have revised the label to: "Number of perfluorinated carbon units". The revised Figure 3 is shown below (on page 8):

[Figure]

**Figure 3.** The impacts of perfluorinated carbon length on the enthalpies of activation at 0 K in the $C_nH_{2n+1}CHO/C_nF_{2n+1}CHO + HO_2$ reactions. The values for $C_nH_{2n+1}CHO$ (n = 1-5) + $HO_2$ and $C_nF_{2n+1}CHO + HO_2$ are obtained from references (Ding and Long, 2022; Gao et al., 2024) and calculated by using FNO-CCSD(T)-F12a/cc-pVDZ-F12.

**(d) Comment:** The calculated high-pressure limit rate constants (e.g., $k_1=5.42\times10^{-14}$-$3.35\times10^{-12}$ cm³ molecule$^{-1}$ s$^{-1}$) lack comparison with experimental data or analogous systems (e.g., non-fluorinated aldehydes + $HO_2$), reducing confidence in the results. (lines 204-207)

**Response:** The revised manuscript now includes a comparison of the calculated rate constants for fluorinated aldehydes ($C_2F_5CHO$ and $C_3F_7CHO$) with experimental and theoretical data from non-fluorinated aldehydes ($C_2H5CHO$ and $C_3H_7CHO$). This comparison shows that the rate constants for both fluorinated and non-fluorinated aldehydes exhibit similar magnitudes and temperature dependencies, which strengthens the confidence in the calculated results. The revised text is highlighted in the manuscript for easy reference.

Revised to: "Regarding the $C_2F_5CHO+ HO_2$ reaction, the rate constant $k_1$ exhibits a decrease from $3.35 \times 10^{-12}$ cm³ molecule$^{-1}$ s$^{-1}$ at 190 K to $5.42 \times 10^{-14}$ cm³ molecule$^{-1}$ s$^{-1}$ at 350 K in Figure 4 and Tables S5-S7. Similarly, the rate constant $k_2$ for the $C_3F_7CHO + HO_2$ reaction also decreases with increasing temperature. These trends are consistent with theoretical studies of non-fluorinated aldehydes such as $C_2H_5CHO$ and $C_3H_7CHO$, where

rate constants for reactions with HO$_2$ were reported in the range of $10^{-14}$ to $10^{-13}$ cm$^3$ molecule$^{-1}$ s$^{-1}$ at atmospheric temperatures, indicating similar reactivity between fluorinated and non-fluorinated aldehydes with HO$_2$.(Ding and Long, 2022; Gao et al., 2024)" on page 9

**(e) Comment:** There is inconsistency in the units used for atmospheric lifetimes. Table 2 reports lifetimes in seconds, while the discussion section uses hours. The authors should standardize the units throughout the manuscript to avoid confusion. (lines 250-255)

**Response:** We normalized all lifetime values to hours in the revised manuscript and marked them in the header of Table 2. (on page 13)

**Table 2.** Hydroperoxyl radical concentration (in molecules cm$^{-3}$), rate constants (in cm$^3$ molecule$^{-1}$ s$^{-1}$), and atmospheric lifetimes (in hour) with respect to bimolecular reactions as functions of altitude.

| H (km)[a] | T (K)[a] | P (mbar)[a] | [HO$_2$][b] | $k_1$ (T, p)[c] | $k_2$ (T, p)[c] | $\tau_1$[d] | $\tau_2$[d] |
|---|---|---|---|---|---|---|---|
| 0 | 290.2 | 1010 | 1.40E+08 | 1.38E-13 | 9.18E-14 | 1.44E+01 | 2.16E+01 |
| 5 | 250.5 | 496 | 4.90E+07 | 3.44E-13 | 2.30E-13 | 1.65E+01 | 2.47E+01 |
| 10 | 215.6 | 243 | 8.30E+06 | 1.07E-12 | 6.47E-13 | 3.13E+01 | 5.18E+01 |
| 15 | 198 | 119 | 2.30E+06 | 2.21E-12 | 2.98E-13 | 5.47E+01 | 4.05E+02 |
| 20 | 208 | 58.2 | 2.90E+06 | 1.38E-12 | 1.22E-13 | 6.93E+01 | 7.85E+02 |
| 25 | 216.1 | 28.5 | 5.70E+06 | 9.38E-13 | 4.32E-14 | 5.20E+01 | 1.13E+03 |
| 30 | 221.5 | 13.9 | 7.50E+06 | 6.52E-13 | 1.29E-14 | 5.68E+01 | 2.88E+03 |
| 35 | 228.1 | 6.83 | 6.90E+06 | 4.04E-13 | 4.13E-15 | 9.96E+01 | 9.74E+03 |
| 40 | 240.5 | 3.34 | 5.90E+06 | 2.29E-13 | 1.73E-15 | 2.06E+02 | 2.73E+04 |
| 45 | 251.9 | 1.64 | 4.90E+06 | 1.27E-13 | 6.56E-16 | 4.45E+02 | 8.64E+04 |
| 50 | 253.7 | 0.801 | 4.00E+06 | 5.87E-14 | 1.80E-16 | 1.18E+03 | 3.86E+05 |

[a]H denotes altitude (atmospheric scale height); T denotes temperature; p denotes pressure.
[b]Data are from ref (Brasseur and Solomon, 2006). [c]$k_1$, $k_2$ are the rate constants of the HO$_2$ reactions with C$_2$F$_5$CHO and C$_3$F$_7$CHO, respectively. [d]$\tau_1$, $\tau_2$ are the atmospheric lifetimes of the HO$_2$ reaction with C$_2$F$_5$CHO and C$_3$F$_7$CHO, respectively

**5. Atmospheric Implications**

**(a) Comment:** The GEOS-Chem simulation results (Lines 273–287) focus on HO$_2$/OH ratios but do not discuss diurnal variation, which could affect the dominance of HO$_2$ pathways.

**Response:** Thank you for your constructive comments. We have added a distribution map of the HO$_2$/OH ratio during the day (see Figure S4) and conducted a detailed analysis.

we have added an additional paragraph in the revised manuscript: "Our further analysis of the global distribution of the ratio between HO$_2$ and OH during daytime reveals that HO$_2$ concentrations are generally higher than OH concentrations (see Figure S4). Notably, along the west coast of South America (approximately between 0° and 30°S latitude and 60°W to 120°W longitude), the ratio can reach up to three orders of magnitude. Industrial areas (such as Russia and Malaysia) and certain regions in Africa also exhibit high ratios of 1-2 orders of magnitude. This suggests that in these areas, the concentration of HO$_2$ is significantly higher than that of OH, which may be related to local industrial activities or specific emission characteristics. However, due to the presence of daytime photolysis, the

generation and loss pathways of $HO_2$ and OH become more complex, leading to significant uncertainty in interpreting the ratio. For instance, photolysis can alter the formation rates of $HO_2$ and OH, thereby affecting their concentration ratio. Additionally, the high ratios along the eastern coast of North America may be associated with atmospheric transport and regional emission features. Despite these complexities, the high daytime ratios still indicate that in some regions, $HO_2$ may play a role in the oxidation pathways of $C_2F_5CHO$ and $C_3H_7CHO$ both during the day and at night." On page 14

[Figure]

Figure S4. The annual average ratio of $HO_2$/OH during the day globally.

**(b) Comment:**

The prospect for future research is somewhat brief and doesn't adequately take into account the current study's limitations and possible areas for expansion. (lines 321-325)

**Response:** Thanks to the reviewer, we expanded the discussion to highlight limitations and future directions: "While the present investigation establishes the $HO_2$-mediated degradation pathway for linear perfluoroaldehydes ($C_2F_5CHO$/$C_3F_7CHO$), it simultaneously highlights critical gaps in our understanding of their atmospheric lifetimes. Notably, the current work focuses on gas-phase $HO_2$ reactions. However, the roles of heterogeneous interfacial processes (e.g., on aerosol surfaces or cloud droplets) remains unexplored.(Zhang et al., 2024) The potential for $HO_2$-driven defluorination to generate reactive $CF_3$ radicals, which could initiate secondary reactions (e.g., with $O_3$ or $NO_2$), requires systematic investigation to assess implications for atmospheric oxidizing capacity and secondary aerosol formation. Additionally, the study focuses on radical-driven pathways but acknowledges that photolysis is a competing sink for linear perfluoroaldehydes. Future work should quantify photolysis rates under stratospheric UV conditions (e.g., 200–300 nm) to reconcile discrepancies between modeled and observed atmospheric lifetimes.(Thomson et al., 2025) Addressing these limitations will require integrating advanced experimental techniques (e.g., synchrotron-based photoionization mass spectrometry) with multi-scale modeling frameworks, while prioritizing under sampled environments like the upper troposphere and polar regions where $HO_2$ reactivity

anomalies could profoundly alter PFAS degradation trajectories.(Alam et al., 2024; Zhou et al., 2024) Such efforts are critical for refining environmental risk assessments of emerging HFOs and guiding the design of next-generation chemicals with minimized atmospheric persistence." On page 16

**6. References**
**(a) Comment:**
Some of the cited references are incomplete or incorrect.

Lee, B. H., Munger, J. W., Wofsy, S. C., Rizzo, L. V., Yoon, J. Y. S., Turner, A. J., Thornton, J. A., and Swann, A. L. S.: Sensitive Response of Atmospheric Oxidative Capacity to the Uncertainty in the Emissions of Nitric Oxide (NO) From Soils 450 in Amazonia, Geophysical Research Letters, 51, 1-10, https://doi.org/10.1029/2023GL107214, 2024. (lines 448-450)

Long, B., Bao, J. L., and Truhlar, D. G.: Rapid unimolecular reaction of stabilized Criegee intermediates and implications for atmospheric chemistry, Nature Communications, 10, 1–8, https://doi.org/10.1038/s41467-019-09948-7, 2019. (lines 473-474)

Xia, D., Zhang, H., Ju, Y., Xie, H., Su, L., Ma, F., Jiang, J., Chen, J., and Francisco, J. S.: Spontaneous Degradation of the "Forever Chemicals" Perfluoroalkyl and Polyfluoroalkyl Substances (PFASs) on Water Droplet Surfaces, https://doi.org/10.1021/jacs.4c00435, 2024. (lines 548-550)

**Response:**
All references have been reformatted to comply with journal guidelines (AGU style) and updated with missing details:

Lee, B. H., Munger, J. W., Wofsy, S. C., Rizzo, L. V., Yoon, J. Y. S., Turner, A. J., Thornton, J. A., and Swann, A. L. S.: Sensitive Response of Atmospheric Oxidative Capacity to the Uncertainty in the Emissions of Nitric Oxide (NO) From Soils 450 in Amazonia, Geophysical Research Letters, 51, e2023GL107214, https://doi.org/10.1029/2023GL107214, 2024. (lines 519-521)

Long, B., Bao, J. L., and Truhlar, D. G.: Rapid unimolecular reaction of stabilized Criegee intermediates and implications for atmospheric chemistry, Nature Communications, 10, 2003, https://doi.org/10.1038/s41467-019-09948-7, 2019. (lines 547-548)

Xia, D., Zhang, H., Ju, Y., Xie, H., Su, L., Ma, F., Jiang, J., Chen, J., and Francisco, J. S.: Spontaneous Degradation of the "Forever Chemicals" Perfluoroalkyl and Polyfluoroalkyl Substances (PFASs) on Water Droplet Surfaces, Journal of the American Chemical Society, 146, 11266–11271, https://doi.org/10.1021/jacs.4c00435, 2024. (lines 631-633)

**References (in revised manuscript)**

Alam, M. S., Abbasi, A., and Chen, G.: Fate, distribution, and transport dynamics of Per- and Polyfluoroalkyl Substances (PFASs) in the environment, Journal of Environmental Management, 371, 123163, https://doi.org/10.1016/j.jenvman.2024.123163, 2024.

Andersen, M. P. S., Nielsen, O. J., Hurley, M. D., Ball, J. C., Wallington, T. J., Stevens, J. E., Martin, J. W., Ellis, D. A., and Mabury, S. A.: Atmospheric chemistry of n-$C_xF_{2x+1}CHO$ (x = 1, 3, 4): Reaction with Cl atoms, OH radicals and IR spectra of $C_xF$

$_{2x+1}$C(O)O$_2$NO$_2$, Journal of Physical Chemistry A, 108, 5189–5196, https://doi.org/10.1021/jp0496598, 2004.

Brasseur, G. and Solomon, S.: Aeronomy of the middle atmosphere: chemistry and physics of the stratosphere and mesosphere., Springer Science & Business Media, 617–627 pp., https://doi.org/10.1007/1-4020-3824-0, 2006.

Burkholder, J. B., Cox, R. A., and Ravishankara, A. R.: Atmospheric Degradation of Ozone Depleting Substances, Their Substitutes, and Related Species, Chemical Reviews, 115, 3704–3759, https://doi.org/10.1021/cr5006759, 2015.

Chiappero, M. S., Malanca, F. E., Argüello, G. A., Wooldridge, S. T., Hurley, M. D., Ball, J. C., Wallington, T. J., Waterland, R. L., and Buck, R. C.: Atmospheric chemistry of perfluoroaldehydes (C$_x$F$_{2x+1}$CHO) and fluorotelomer aldehydes (C$_x$F$_{2x+1}$CH$_2$CHO): Quantification of the important role, of photolysis, Journal of Physical Chemistry A, 110, 11944–11953, https://doi.org/10.1021/jp064262k, 2006.

Chiappero, M. S., Argüello, G. A., Hurley, M. D., and Wallington, T. J.: Atmospheric chemistry of n-C$_6$F$_{13}$CH$_2$CHO: Formation from n-C$_6$F$_{13}$CH$_2$CH$_2$OH, kinetics, and mechanisms of reactions with chlorine atoms and OH radicals, Journal of Physical Chemistry A, 114, 6131–6137, https://doi.org/10.1021/jp101587m, 2010.

David, L. M., Barth, M., Höglund-Isaksson, L., Purohit, P., Velders, G. J. M., Glaser, S., and Ravishankara, A. R.: Trifluoroacetic acid deposition from emissions of HFO-1234yf in India, China, and the Middle East, Atmospheric Chemistry and Physics, 21, 14833–14849, https://doi.org/10.5194/acp-21-14833-2021, 2021.

Ding, D. P. and Long, B.: Reaction between propionaldehyde and hydroxyperoxy radical in the atmosphere: A reaction route for the sink of propionaldehyde and the formation of formic acid, Atmospheric Environment, 284, 119202, https://doi.org/10.1016/j.atmosenv.2022.119202, 2022.

Dong, Z. G., Xu, F., Mitchell, E., and Long, B.: Trifluoroacetaldehyde aminolysis catalyzed by a single water molecule: An important sink pathway for trifluoroacetaldehyde and a potential pathway for secondary organic aerosol growth, Atmospheric Environment, 249, 118242, https://doi.org/10.1016/j.atmosenv.2021.118242, 2021.

Gao, Q., Shen, C., Zhang, H., Long, B., and Truhlar, D. G.: Quantitative Kinetics Reveal that Reactions of HO$_2$ are a Significant Sink for Aldehydes in the Atmosphere and may Initiate the Formation of Highly Oxygenated Molecules via Autoxidation , Physical Chemistry Chemical Physics, 26, 16160–16174, https://doi.org/10.1039/d4cp00693c, 2024.

Kelly, T., Sidebottom, H., Nielsen, C. J., and Sellevag, S. R.: A study of the IR and UV-Vis absorption cross-sections, photolysis and OH-initiated oxidation of CF$_3$CHO and CF$_3$CH$_2$CHO, Physical Chemistry Chemical Physics, 1243–1252, 2004.

Long, B., Xia, Y., and Truhlar, D. G.: Quantitative Kinetics of HO$_2$ Reactions with Aldehydes in the Atmosphere: High-Order Dynamic Correlation, Anharmonicity, and Falloff Effects Are All Important, Journal of the American Chemical Society, 144, 19910–19920, https://doi.org/10.1021/jacs.2c07994, 2022.

Thomson, J. D., Campbell, J. S., Edwards, E. B., Medcraft, C., Nauta, K., Pérez-Peña, M. P., Fisher, J. A., Osborn, D. L., Kable, S. H., and Hansen, C. S.: Fluoroform (CHF

3 ) Production from CF$_3$CHO Photolysis and Implications for the Decomposition of Hydrofluoroolefins and Hydrochlorofluoroolefins in the Atmosphere, Journal of the American Chemical Society, 147, 33–38, https://doi.org/10.1021/jacs.4c11776, 2025.

Wang, Y., Liu, J. yao, Yang, L., Zhao, X. lei, Ji, Y. M., and Li, Z. sheng: Theoretical studies and rate constant calculations of the reactions C$_2$F$_5$CHO with OH radicals and Cl atoms, Journal of Molecular Structure: THEOCHEM, 820, 26–34, https://doi.org/10.1016/j.theochem.2007.06.001, 2007.

Wang, Y., Wang, Z., Sun, M., Guo, J., and Zhang, J.: Emissions, degradation and impact of HFO-1234ze from China PU foam industry, Science of the Total Environment, 780, 146631, https://doi.org/10.1016/j.scitotenv.2021.146631, 2021.

Wang, Y., Liu, L., Qiao, X., Sun, M., Guo, J., Zhao, B., and Zhang, J.: Atmospheric fate and impacts of HFO-1234yf from mobile air conditioners in East Asia, Science of the Total Environment, 916, 170137, https://doi.org/10.1016/j.scitotenv.2024.170137, 2024.

Xia, Y., Long, B., Liu, A., and Truhlar, D. G.: Reactions with criegee intermediates are the dominant gas-phase sink for formyl fluoride in the atmosphere, Fundamental Research, 4, 1216–1224, https://doi.org/10.1016/j.fmre.2023.02.012, 2024.

Zhang, W., Issa, K., Tang, T., and Zhang, H.: Role of Hydroperoxyl Radicals in Heterogeneous Oxidation of Oxygenated Organic Aerosols, Environmental Science & Technology, 58, 4727–4736, https://doi.org/10.1021/acs.est.3c09024, 2024.

Zhou, Y., Wang, X., Wang, C., Ji, Z., Niu, X., and Dong, H.: Fate of 'forever chemicals' in the global cryosphere, Earth-Science Reviews, 259, 104973, https://doi.org/10.1016/j.earscirev.2024.104973, 2024.

Ziemann, P. J. and Atkinson, R.: Kinetics, products, and mechanisms of secondary organic aerosol formation, Chemical Society Reviews, 41, 6582–6605, https://doi.org/10.1039/c2cs35122f, 2012.

---

## Author Response (AR1)

Response to the Anonymous Referee #1 Comments for the manuscript "Reaction between perfluoroaldehydes and hydroperoxy radical in the atmosphere: Reaction mechanisms, reaction kinetics modelling, and atmospheric implications"

We sincerely appreciate the time and effort you dedicated to reviewing our manuscript. Your insightful feedback has significantly enhanced both the scientific content and written presentation, elevated the paper's academic value while improved its accessibility and clarity for readers. Below, we provide point-by-point responses to your comments. All corresponding revisions are marked in the tracked-changes version of the updated manuscript, where reviewer comments (RC) appear in black text and author responses (AC) in blue.

**Specific Comments**

**1. Abstract**

(a) Comment: the atmospheric lifetimes for C2F5CHO and C3F7CHO, approximately 14.4-31.3 hours and 21.6-51.8 hours by HO2 are much shorter than those via OH radical," it is not validated. (lines 15-16)

**Response:** We thank the reviewer for highlighting the need for validation of the atmospheric lifetime. In the revised manuscript, we define the atmospheric lifetime  $\tau$  for linear perfluorinated aldehydes reacting with HO2 as  $\tau = 1/(k[\text{HO}_2])$ , where k is the bimolecular rate constant and [HO2] represents the concentration of HO2 radicals. The full atmospheric lifetime data are presented in Section 3.3, where atmospheric lifetimes with respect to bimolecular reactions as functions of altitude are list in Table 2.

We revised the sentence " $\tau_1$ , and  $\tau_2$  are the atmospheric lifetimes of the HO2 reaction with C2F5CHO and C3F7CHO, respectively." to " $\tau_1 = 1/(k_1[\text{HO}_2])$  and  $\tau_2 = 1/(k_2[\text{HO}_2])$  define the atmospheric lifetimes for HO2 reaction with C2F5CHO and C3F7CHO, respectively." on page 13

We also revised the sentence "As illustrated in Table 2, the HO2-mediated elimination pathways for  $C_2F_5CHO$  and  $C_3F_7CHO$  are characterized by relatively rapid reaction rates within the troposphere (< 10 km), resulting in relatively short atmospheric lifetimes of approximately  $5.18 \times 10^4$  -  $1.13 \times 10^5$  s (14.4 - 31.3 hours) and  $7.78 \times 10^4$  -  $1.86 \times 10^5$  s (21.6 - 51.8 hours)." to "The HO2-mediated elimination pathways for  $C_2F_5CHO$  and  $C_3F_7CHO$  are characterized by relatively rapid reaction rates within the troposphere (< 10 km), resulting in relatively short atmospheric lifetimes of approximately 14.4 - 31.3 hours and 21.6 - 51.8 hours (See Table 2)." on page 12

**2. Introduction**

(a) Comment: Alternating use of "perfluoroaldehydes" (in tittle) and "linear perfluoroaldehydes" may confuse readers.

**Response:** We have standardized the terminology to "linear perfluoroaldehydes" throughout the manuscript to avoid ambiguity.

**(b)** Comment: The transition from PFAS's GWP to perfluoroaldehyde sources (lines 25-30) is unclear.

**Response:** We added a transitional sentence to clarify the link between PFASs' environmental impact and the formation of linear perfluoroaldehydes: "During their degradation in the atmosphere, PFASs undergo complex chemical transformations, leading to the formation of linear perfluoroaldehydes. (Alam et al., 2024; Burkholder et al., 2015; David et al., 2021; Wang et al., 2021, 2024)" on page 1

(c) Comment: The literature review focuses solely on OH and Cl radicals, omitting potential roles of other oxidants (e.g., O3, NO3). (lines 35-49)

**Response:** We expanded the discussion to acknowledge other oxidants: "Linear perfluoroaldehydes were generally considered to be removed through photochemical reactions (Chiappero et al., 2006; Kelly et al., 2004) and free radical reactions initiated by OH and Cl radicals (Andersen et al., 2004; Chiappero et al., 2010; Wang et al., 2007). Additionally, NO3 may also contribute to their atmospheric degradation. (Burkholder et al., 2015; Ziemann and Atkinson, 2012)" on page 2

(d) Comment: The long sentence "Moreover, the rate constant of Cl atoms with  $C_nF_{2n+1}CHO$  (1,2, 3, 4) is around  $2\times 10^{-12}\, cm^3$  molecule  $^{-1}s^{-1}$ , which is slightly faster than that of the OH radical reactions." (lines 46-47) has poor readability and unclear notes.

**Response:** We split the sentence for clarity: "Moreover, the rate constant of Cl atoms with  $C_nF_{2n+1}$ CHO (n = 1-4) is approximately  $2 \times 10^{-12}$  cm3 molecule-1 s-1. This value is slightly faster than that of the corresponding OH radical reactions under similar conditions." on page 2

**3. Computational Methods**

(a) Comment: In section 2.1, the "CCSD(T)-F12a/cc-pVTZ-F12" method is mentioned, but the text fails to demonstrating its applicability to perfluorinated compound systems. (lines 75-78)

**Response:** We added validation references for fluorinated systems: "Furthermore, CCSD(T)-F12a/cc-pVTZ-F12 has been shown good performance for molecules containing fluorine atoms.(Dong et al., 2021; Long et al., 2022; Xia et al., 2024)" on page 3

**4. Results and Discussion**

(a) Comment: Figure 1 contains no relevant information. "The result shows that the specific reaction scale factors are 0.955 for TS1 (See Figure 1) and 0.956 for TS2 (See Figure 1)". (lines 95-96)

**Response:** We corrected the text to reference the correct Table (Table S1 in the Supplement) and clarified the context: "The result shows that the specific reaction scale factors are 0.955 for TS1 (see Table S1) and 0.956 for TS2 (see Table S1), which ..." on page 4

**(b)** Comment: Grammatical error in "The torsion of the C-C bond gives produces multiple conformers". (lines 152-153)

**Response:** Corrected to: "Nevertheless, the internal rotation of the C–C bond produces multiple conformers..." on page 6

**(c)** Comment: Figure 3's X-axis label ("Number of alkyl functional groups") is misleading, as the compounds are perfluorinated. (lines 180-185)

**Response:** We appreciate the reviewer's precision feedback and we have revised the label to: "Number of perfluorinated carbon units". The revised Figure 3 is shown below (on page 8):

**Figure 3.** The impacts of perfluorinated carbon length on the enthalpies of activation at 0 K in the  $C_nH_{2n+1}CHO/C_nF_{2n+1}CHO + HO_2$  reactions. The values for  $C_nH_{2n+1}CHO$  (n = 1-5) +  $HO_2$  and  $C_nF_{2n+1}CHO + HO_2$  are obtained from references (Ding and Long, 2022; Gao et al., 2024) and calculated by using FNO-CCSD(T)-F12a/cc-pVDZ-F12.

(d) Comment: The calculated high-pressure limit rate constants (e.g.,  $k_1=5.42\times10^{-14}$ -  $3.35\times10^{-12}$  cm3 molecule-1 s-1) lack comparison with experimental data or analogous systems (e.g., non-fluorinated aldehydes + HO2), reducing confidence in the results. (lines 204-207)

**Response:** The revised manuscript now includes a comparison of the calculated rate constants for fluorinated aldehydes (C2F5CHO and C3F7CHO) with experimental and theoretical data from non-fluorinated aldehydes (C2H5CHO and C3H7CHO). This comparison shows that the rate constants for both fluorinated and non-fluorinated aldehydes exhibit similar magnitudes and temperature dependencies, which strengthens the confidence in the calculated results. The revised text is highlighted in the manuscript for easy reference.

Revised to: "Regarding the  $C_2F_5CHO+ HO_2$  reaction, the rate constant  $k_1$  exhibits a decrease from  $3.35 \times 10^{-12}$  cm3 molecule-1 s-1 at 190 K to  $5.42 \times 10^{-14}$  cm3 molecule-1 s-1 at 350 K in Figure 4 and Tables S5-S7. Similarly, the rate constant  $k_2$  for the  $C_3F_7CHO + HO_2$  reaction also decreases with increasing temperature. These trends are consistent with theoretical studies of non-fluorinated aldehydes such as  $C_2H_5CHO$  and  $C_3H_7CHO$ , where

rate constants for reactions with HO2 were reported in the range of 10-14 to 10-13 cm3 molecule-1 s-1 at atmospheric temperatures, indicating similar reactivity between fluorinated and non-fluorinated aldehydes with HO2.(Ding and Long, 2022; Gao et al., 2024)" on page 9

**(e)** Comment: There is inconsistency in the units used for atmospheric lifetimes. Table 2 reports lifetimes in seconds, while the discussion section uses hours. The authors should standardize the units throughout the manuscript to avoid confusion. (lines 250-255)

**Response:** We normalized all lifetime values to hours in the revised manuscript and marked them in the header of Table 2. (on page 13)

**Table 2.** Hydroperoxyl radical concentration (in molecules cm-3), rate constants (in cm3 molecule-1 s -1), and atmospheric lifetimes (in hour) with respect to bimolecular reactions as functions of altitude.

| H (km) a | T (K) a | P (mbar) a | $[HO_2]^b$ | $k_1(T, p)^c$ | $k_2(T, p)^c$ | ${\color{red}	au_1}^{ m d}$ | ${	au_2}^{ m d}$ |
|---------------------|--------------------|-----------------------|------------|---------------|---------------|-----------------------------|------------------|
| 0                   | 290.2              | 1010                  | 1.40E+08   | 1.38E-13      | 9.18E-14      | 1.44E+01                    | 2.16E + 01       |
| 5                   | 250.5              | 496                   | 4.90E+07   | 3.44E-13      | 2.30E-13      | 1.65E+01                    | 2.47E + 01       |
| 10                  | 215.6              | 243                   | 8.30E+06   | 1.07E-12      | 6.47E-13      | 3.13E+01                    | 5.18E+01         |
| 15                  | 198                | 119                   | 2.30E+06   | 2.21E-12      | 2.98E-13      | 5.47E+01                    | 4.05E+02         |
| 20                  | 208                | 58.2                  | 2.90E+06   | 1.38E-12      | 1.22E-13      | 6.93E+01                    | 7.85E+02         |
| 25                  | 216.1              | 28.5                  | 5.70E+06   | 9.38E-13      | 4.32E-14      | 5.20E+01                    | 1.13E+03         |
| 30                  | 221.5              | 13.9                  | 7.50E+06   | 6.52E-13      | 1.29E-14      | 5.68E+01                    | 2.88E + 03       |
| 35                  | 228.1              | 6.83                  | 6.90E+06   | 4.04E-13      | 4.13E-15      | 9.96E+01                    | 9.74E+03         |
| 40                  | 240.5              | 3.34                  | 5.90E+06   | 2.29E-13      | 1.73E-15      | 2.06E + 02                  | 2.73E+04         |
| 45                  | 251.9              | 1.64                  | 4.90E+06   | 1.27E-13      | 6.56E-16      | 4.45E+02                    | 8.64E+04         |
| 50                  | 253.7              | 0.801                 | 4.00E+06   | 5.87E-14      | 1.80E-16      | 1.18E + 03                  | 3.86E+05         |

<sup>aH denotes altitude (atmospheric scale height); T denotes temperature; p denotes pressure. bData are from ref (Brasseur and Solomon, 2006).  $^ck_1$ ,  $k_2$  are the rate constants of the HO2 reactions with C2F5CHO and C3F7CHO, respectively.  $^d\tau_1$ ,  $\tau_2$  are the atmospheric lifetimes of the HO2 reaction with C2F5CHO and C3F7CHO, respectively

**5. Atmospheric Implications**

(a) Comment: The GEOS-Chem simulation results (Lines 273–287) focus on HO2/OH ratios but do not discuss diurnal variation, which could affect the dominance of HO2 pathways.

Response: Thank you for your constructive comments. We have added a distribution map of the HO2/OH ratio during the day (see Figure S4) and conducted a detailed analysis. we have added an additional paragraph in the revised manuscript: "Our further analysis of the global distribution of the ratio between HO2 and OH during daytime reveals that HO2 concentrations are generally higher than OH concentrations (see Figure S4). Notably, along the west coast of South America (approximately between 0° and 30°S latitude and 60°W to 120°W longitude), the ratio can reach up to three orders of magnitude. Industrial areas (such as Russia and Malaysia) and certain regions in Africa also exhibit high ratios of 1-2 orders of magnitude. This suggests that in these areas, the concentration of HO2 is significantly higher than that of OH, which may be related to local industrial activities or specific emission characteristics. However, due to the presence of daytime photolysis, the

generation and loss pathways of HO2 and OH become more complex, leading to significant uncertainty in interpreting the ratio. For instance, photolysis can alter the formation rates of HO2 and OH, thereby affecting their concentration ratio. Additionally, the high ratios along the eastern coast of North America may be associated with atmospheric transport and regional emission features. Despite these complexities, the high daytime ratios still indicate that in some regions, HO2 may play a role in the oxidation pathways of C2F5CHO and C3H7CHO both during the day and at night." On page 14

Figure S4. The annual average ratio of HO2/OH during the day globally.

**(b) Comment:**

The prospect for future research is somewhat brief and doesn't adequately take into account the current study's limitations and possible areas for expansion. (lines 321-325)

**Response:** Thanks to the reviewer, we expanded the discussion to highlight limitations and future directions: "While the present investigation establishes the HO2-mediated degradation pathway for linear perfluoroaldehydes (C2F5CHO/C3F7CHO), it simultaneously highlights critical gaps in our understanding of their atmospheric lifetimes. Notably, the current work focuses on gas-phase HO2 reactions. However, the roles of heterogeneous interfacial processes (e.g., on aerosol surfaces or cloud droplets) remains unexplored.(Zhang et al., 2024) The potential for HO2-driven defluorination to generate reactive CF3 radicals, which could initiate secondary reactions (e.g., with O3 or NO2), requires systematic investigation to assess implications for atmospheric oxidizing capacity and secondary aerosol formation. Additionally, the study focuses on radical-driven pathways but acknowledges that photolysis is a competing sink for linear perfluoroaldehydes. Future work should quantify photolysis rates under stratospheric UV conditions (e.g., 200-300 nm) to reconcile discrepancies between modeled and observed atmospheric lifetimes. (Thomson et al., 2025) Addressing these limitations will require integrating advanced experimental techniques (e.g., synchrotron-based photoionization mass spectrometry) with multi-scale modeling frameworks, while prioritizing under sampled environments like the upper troposphere and polar regions where HO2 reactivity

anomalies could profoundly alter PFAS degradation trajectories. (Alam et al., 2024; Zhou et al., 2024) Such efforts are critical for refining environmental risk assessments of emerging HFOs and guiding the design of next-generation chemicals with minimized atmospheric persistence." On page 16

**6. References**

**(a) Comment:**

Some of the cited references are incomplete or incorrect.

Lee, B. H., Munger, J. W., Wofsy, S. C., Rizzo, L. V., Yoon, J. Y. S., Turner, A. J., Thornton, J. A., and Swann, A. L. S.: Sensitive Response of Atmospheric Oxidative Capacity to the Uncertainty in the Emissions of Nitric Oxide (NO) From Soils 450 in Amazonia, Geophysical Research Letters, 51, 1-10, https://doi.org/10.1029/2023GL107214, 2024. (lines 448-450)

Long, B., Bao, J. L., and Truhlar, D. G.: Rapid unimolecular reaction of stabilized Criegee intermediates and implications for atmospheric chemistry, Nature Communications, 10, 1–8, https://doi.org/10.1038/s41467-019-09948-7, 2019. (lines 473-474)

Xia, D., Zhang, H., Ju, Y., Xie, H., Su, L., Ma, F., Jiang, J., Chen, J., and Francisco, J. S.: Spontaneous Degradation of the "Forever Chemicals" Perfluoroalkyl and Polyfluoroalkyl Substances (PFASs) on Water Droplet Surfaces, https://doi.org/10.1021/jacs.4c00435, 2024. (lines 548-550)

**Response:**

All references have been reformatted to comply with journal guidelines (AGU style) and updated with missing details:

Lee, B. H., Munger, J. W., Wofsy, S. C., Rizzo, L. V., Yoon, J. Y. S., Turner, A. J., Thornton, J. A., and Swann, A. L. S.: Sensitive Response of Atmospheric Oxidative Capacity to the Uncertainty in the Emissions of Nitric Oxide (NO) From Soils 450 in Amazonia, Geophysical Research Letters, 51, e2023GL107214, https://doi.org/10.1029/2023GL107214, 2024. (lines 519-521)

Long, B., Bao, J. L., and Truhlar, D. G.: Rapid unimolecular reaction of stabilized Criegee intermediates and implications for atmospheric chemistry, Nature Communications, 10, 2003, https://doi.org/10.1038/s41467-019-09948-7, 2019. (lines 547-548)

Xia, D., Zhang, H., Ju, Y., Xie, H., Su, L., Ma, F., Jiang, J., Chen, J., and Francisco, J. S.: Spontaneous Degradation of the "Forever Chemicals" Perfluoroalkyl and Polyfluoroalkyl Substances (PFASs) on Water Droplet Surfaces, Journal of the American Chemical Society, 146, 11266–11271, https://doi.org/10.1021/jacs.4c00435, 2024. (lines 631-633)

**References (in revised manuscript)**

Alam, M. S., Abbasi, A., and Chen, G.: Fate, distribution, and transport dynamics of Per- and Polyfluoroalkyl Substances (PFASs) in the environment, Journal of Environmental Management, 371, 123163, https://doi.org/10.1016/j.jenvman.2024.123163, 2024.

Andersen, M. P. S., Nielsen, O. J., Hurley, M. D., Ball, J. C., Wallington, T. J., Stevens, J. E., Martin, J. W., Ellis, D. A., and Mabury, S. A.: Atmospheric chemistry of  $n-C_xF$   $_{2x+1}CHO$  (x=1,3,4): Reaction with Cl atoms, OH radicals and IR spectra of  $C_xF$

- 2x+1C(O)O2NO2, Journal of Physical Chemistry A, 108, 5189–5196, https://doi.org/10.1021/jp0496598, 2004.
- Brasseur, G. and Solomon, S.: Aeronomy of the middle atmosphere: chemistry and physics of the stratosphere and mesosphere., Springer Science & Business Media, 617–627 pp., https://doi.org/10.1007/1-4020-3824-0, 2006.
- Burkholder, J. B., Cox, R. A., and Ravishankara, A. R.: Atmospheric Degradation of Ozone Depleting Substances, Their Substitutes, and Related Species, Chemical Reviews, 115, 3704–3759, https://doi.org/10.1021/cr5006759, 2015.
- Chiappero, M. S., Malanca, F. E., Argüello, G. A., Wooldridge, S. T., Hurley, M. D., Ball, J. C., Wallington, T. J., Waterland, R. L., and Buck, R. C.: Atmospheric chemistry of perfluoroaldehydes ( $C_xF_{2x+1}CHO$ ) and fluorotelomer aldehydes ( $C_xF_{2x+1}CH_2CHO$ ): Quantification of the important role, of photolysis, Journal of Physical Chemistry A, 110, 11944–11953, https://doi.org/10.1021/jp064262k, 2006.
- Chiappero, M. S., Argüello, G. A., Hurley, M. D., and Wallington, T. J.: Atmospheric chemistry of n-C6F13CH2CHO: Formation from n-C6F13CH2CH2OH, kinetics, and mechanisms of reactions with chlorine atoms and OH radicals, Journal of Physical Chemistry A, 114, 6131–6137, https://doi.org/10.1021/jp101587m, 2010.
- David, L. M., Barth, M., Höglund-Isaksson, L., Purohit, P., Velders, G. J. M., Glaser, S., and Ravishankara, A. R.: Trifluoroacetic acid deposition from emissions of HFO-1234yf in India, China, and the Middle East, Atmospheric Chemistry and Physics, 21, 14833–14849, https://doi.org/10.5194/acp-21-14833-2021, 2021.
- Ding, D. P. and Long, B.: Reaction between propional dehyde and hydroxyperoxy radical in the atmosphere: A reaction route for the sink of propional dehyde and the formation of formic acid, Atmospheric Environment, 284, 119202, https://doi.org/10.1016/j.atmosenv.2022.119202, 2022.
- Dong, Z. G., Xu, F., Mitchell, E., and Long, B.: Trifluoroacetaldehyde aminolysis catalyzed by a single water molecule: An important sink pathway for trifluoroacetaldehyde and a potential pathway for secondary organic aerosol growth, Atmospheric Environment, 249, 118242, https://doi.org/10.1016/j.atmosenv.2021.118242, 2021.
- Gao, Q., Shen, C., Zhang, H., Long, B., and Truhlar, D. G.: Quantitative Kinetics Reveal that Reactions of HO2 are a Significant Sink for Aldehydes in the Atmosphere and may Initiate the Formation of Highly Oxygenated Molecules via Autoxidation, Physical Chemistry Chemical Physics, 26, 16160–16174, https://doi.org/10.1039/d4cp00693c, 2024.
- Kelly, T., Sidebottom, H., Nielsen, C. J., and Sellevag, S. R.: A study of the IR and UV-Vis absorption cross-sections, photolysis and OH-initiated oxidation of CF3CHO and CF3CH2CHO, Physical Chemistry Chemical Physics, 1243–1252, 2004.
- Long, B., Xia, Y., and Truhlar, D. G.: Quantitative Kinetics of HO2 Reactions with Aldehydes in the Atmosphere: High-Order Dynamic Correlation, Anharmonicity, and Falloff Effects Are All Important, Journal of the American Chemical Society, 144, 19910–19920, https://doi.org/10.1021/jacs.2c07994, 2022.
- Thomson, J. D., Campbell, J. S., Edwards, E. B., Medcraft, C., Nauta, K., Pérez-Peña, M. P., Fisher, J. A., Osborn, D. L., Kable, S. H., and Hansen, C. S.: Fluoroform (CHF

3 ) Production from CF3CHO Photolysis and Implications for the Decomposition of Hydrofluoroolefins and Hydrochlorofluoroolefins in the Atmosphere, Journal of the American Chemical Society, 147, 33–38, https://doi.org/10.1021/jacs.4c11776, 2025. Wang, Y., Liu, J. yao, Yang, L., Zhao, X. lei, Ji, Y. M., and Li, Z. sheng: Theoretical studies and rate constant calculations of the reactions C2F5CHO with OH radicals and Cl atoms, Journal of Molecular Structure: THEOCHEM, 820, 26–34, https://doi.org/10.1016/j.theochem.2007.06.001, 2007.

Wang, Y., Wang, Z., Sun, M., Guo, J., and Zhang, J.: Emissions, degradation and impact of HFO-1234ze from China PU foam industry, Science of the Total Environment, 780, 146631, https://doi.org/10.1016/j.scitotenv.2021.146631, 2021.

Wang, Y., Liu, L., Qiao, X., Sun, M., Guo, J., Zhao, B., and Zhang, J.: Atmospheric fate and impacts of HFO-1234yf from mobile air conditioners in East Asia, Science of the Total Environment, 916, 170137, https://doi.org/10.1016/j.scitotenv.2024.170137, 2024.

Xia, Y., Long, B., Liu, A., and Truhlar, D. G.: Reactions with criegee intermediates are the dominant gas-phase sink for formyl fluoride in the atmosphere, Fundamental Research, 4, 1216–1224, https://doi.org/10.1016/j.fmre.2023.02.012, 2024.

Zhang, W., Issa, K., Tang, T., and Zhang, H.: Role of Hydroperoxyl Radicals in Heterogeneous Oxidation of Oxygenated Organic Aerosols, Environmental Science & Technology, 58, 4727–4736, https://doi.org/10.1021/acs.est.3c09024, 2024.

Zhou, Y., Wang, X., Wang, C., Ji, Z., Niu, X., and Dong, H.: Fate of 'forever chemicals' in the global cryosphere, Earth-Science Reviews, 259, 104973, https://doi.org/10.1016/j.earscirev.2024.104973, 2024.

Ziemann, P. J. and Atkinson, R.: Kinetics, products, and mechanisms of secondary organic aerosol formation, Chemical Society Reviews, 41, 6582–6605, https://doi.org/10.1039/c2cs35122f, 2012.

Response to the Anonymous Referee #2 Comments for the manuscript "Reaction between perfluoroaldehydes and hydroperoxy radical in the atmosphere: Reaction mechanisms, reaction kinetics modelling, and atmospheric implications"

We sincerely appreciate the time and effort dedicated to reviewing our manuscript. We thank the reviewers for their constructive feedback, which has helped us improve the clarity and impact of our work. Below, we address your concerns and provide a point-by-point response to your comments. Reviewer comments (RC) are in black font and author comments (AC) are in blue.

Zegang Dong and coworkers have carried out an extensive theoretical investigation on the gas phase reaction between two perfluoroaldehydes and hydroperoxyl radical and estimated their atmospheric implications. The work is extensive, the theoretical model is well chosen and already shown to be appropriate for similar class of reactions, the results are properly presented and explained.

**Response:** Thank you for your positive comments.**

However, the work falls severely short in originality and fails to provide any new insight, and is simply an extension of their earlier work (*J. Am. Chem. Soc.* 2022, 144, 19910–19920). Both the reactions studied here are close replica, mechanistically, energetically and kinetically, of the reaction between CF3CHO and HO2 studied in their previous work. As the reactions are always centered at the carbonyl group of reacting aldehydes, with no involvement of the side chains, this is very much expected that a mere elongation of the side chain would not have any dramatic effect on the reactions.

**Response:** Thank you for your comments. We agree with your view that the present work is based on our previous investigations. We do not agree with your view that the elongation of the side chain would not have any dramatic effects on the reactions. In fact, there are two important influences on reaction thermodynamics and kinetics.

First, longer chain for perfluoroaldehydes leads to producing multiple conformers in both reactants and transition states (See Figures 3, S1, and S3), with different energy distributions for these conformers, spanning 0-1.7 and 0-1.9 kcal/mol, respectively. This leads to a reduction in the multi-structure torsional anharmonicity factor ( $F_2^{MS-T}$ ) to 0.45–0.52, compared to 1.0 in one conformer systems in our previous article, resulting in a 50% decrease in rate constants relative to single-conformer C3F7CHO and a 34–43% decrease relative to CF3CHO. This effect has not been observed in our previous study (*J. Am. Chem. Soc.* 2022, *144*, 19910–19920), which cannot be obtained without further investigations. Additionally, for larger molecules such as C4F9CHO and C5F11CHO, the increased molecular size and number of conformers (e.g., 36 conformers in the transition state of C5F11CHO with energy distributions up to 4.8 kcal/mol) further complicate the potential energy surface calculations. These findings highlight that the conformational diversity and molecular size effects significantly alter reaction kinetics, even when the enthalpy of

activation remains relatively insensitive to chain length.

Second,  $C_3F_7CHO + HO_2$  exhibits remarkably pressure dependence, with transition pressures ( $p_{1/2}$ ) ranging from 0.026 to 2.3 bar at 190 – 350 K (See Figure 8, Table S10). This contrasts sharply with  $CF_3CHO + HO_2$ , where pressure dependence is negligible. These findings demonstrate that extending the side chain not only have important influences on reaction kinetics through conformational diversity, but also causes pressure-dependent behavior that cannot be captured by simple empirical models. These findings deepen our understanding of the atmospheric chemistry of polyfluoroalkyl substances (PFAS) and provide critical theoretical foundations for modeling the degradation kinetics of complex PFAS compounds.

Third, this study addresses the challenges on high-accurate quantum chemical calculations for longer linear polyfluorinated aldehydes by developing a computational strategy based on the frozen natural orbital (FNO) approximation, specifically the FNO-CCSD(T)-F12/cc-pVDZ-F12 method. In the present work, we also find that this approach significantly provides computational efficiency and reaches sub-kcal/mol (<1 kcal/mol) accuracy, enabling to predict the enthalpies of activation at 0 K from short-chain to long-chain molecules. Therefore, we believe that the present work has provided new insights on how to obtain the quantitative kinetics of perfluoroaldehydes and hydroperoxy radical.

In our initial manuscript, some key issues may not be clarified; this leads to unclearly revealing the originality and new insights into the present work. Therefore, in the revised article, we have done many corrections to improve the manuscript.

(1) We have rewritten the abstract, "Linear perfluoroaldehydes are important products formed in the atmospheric oxidation of industrial fluorinated compounds. However, their atmospheric lifetimes are incompletely known. Here, we employ high level quantum chemistry methods and a dual-level strategy for kinetics to probe the reactions of C2F5CHO and C3F7CHO with HO2. Our calculated results unveil almost equal activation enthalpies at 0 K for perfluoroaldehyde reaction with HO2, indicating that the carbon chain length minimally influences reaction thermodynamics. Interestingly, the present findings reveal that anharmonicity remarkably enhances the reaction rate constant, whereas multistructural anharmonicity, recrossing, and tunnelling effects exhibit lesser impacts in the C2F5CHO/C3F7CHO + HO2 reaction. In particular, the atmospheric lifetimes for C2F5CHO and C3F7CHO, approximately 14.4-31.3 hours and 21.6-51.8 hours by HO2 are much shorter than those via OH radical, underscoring the dominant removal role of HO2 toward C2F5CHO and C3F7CHO in the atmosphere. Since GEOS-Chem simulation shows that the concentration of HO2 is at least 102 times higher than that of OH radical in Russia, Malaysia, and parts of Africa, the reactions of C2F5CHO and C3F7CHO with HO2 radicals dominate over those with OH radicals and play more vital role in the atmospheric chemical processes of these regions. This study enhances our understanding of the chemical transformations of linear perfluoroaldehydes and provides a scientific foundation for strategies aimed at mitigating their emissions." has been corrected into "Linear perfluoroaldehydes are important products formed in the atmospheric oxidation of industrial fluorinated compounds. However, their atmospheric lifetimes are incompletely known. Here, we employ high level quantum chemistry methods and a dual-level strategy for kinetics to investigate the reactions of  $C_2F_5CHO$  and  $C_3F_7CHO$  with  $HO_2$ . Our calculated results

unveil almost equal activation enthalpies at 0 K for linear perfluoroaldehyde reaction with HO2, indicating that the carbon chain length negligibly influences reaction thermodynamics. The calculated kinetics reveal that vibrational anharmonicity enhance rate constants by a factor of 3–10, while torsional anharmonicity reduces rate constants by 34–55%. Additionally, we also find that the reaction of C3F7CHO with HO2 exhibits significant pressure dependence, with transition pressures ranging from 0.026 to 2.3 bar across a temperature range of 190–350 K. Furthermore, our findings also reveal that the reactions of C2F5CHO and C3F7CHO with HO2 radicals dominate over those with OH radicals in Russia, Malaysia, parts of Africa by the calculated results in combination with data based on global atmospheric chemical model simulations. These findings establish chain-length-dependent pressure effects and conformational sampling as critical, previously unrecognized factors in kinetics calculations, providing a framework for modelling complex fluorotelomer transformations and guiding emission mitigation strategies." on page 1.

(2) In section 1 (Introduction), we have revised the sentence from "Nevertheless, the importance of sink pathway by HO2 is still unknown because there have not been kinetics data for linear perfluoroaldehydes with HO2 in the literature." to "Nevertheless, the importance of sink pathway by HO2 is still unknown because there have not been kinetics data for linear perfluoroaldehydes with HO2 in the literature. Moreover, chain elongation may have influences on reaction kinetics due to multiple conformers. Furthermore, it is unknown for the pressure-dependent effects of larger perfluoroaldehydes with HO2. Although our previous investigations have revealed the importance of CF3CHO + HO2 in the atmosphere (Long et al., 2022), their kinetics of larger perfluoroaldehydes with HO2 are further required to investigate due to the unique features that depend on the specific reaction systems such as multi-structural anharmonicity and pressure effects in these complex systems. Additionally, it is a big challenge for addressing the larger perfluoroaldehydes with HO2 because the computational cost grows very rapidly with system size, making such calculations impractical for high-level quantum chemistry methods." on page 2-3

We also revised the sentence from "These findings could hold significant implications for the formation and yield of HCOOH, thereby contributing to the broader discourse on atmospheric chemistry and environmental policy." to "This study not only resolves the knowledge gap regarding HO2-initiated oxidation of linear perfluoroaldehydes but also establishes a computational strategy for predicting the atmospheric fates of long-chain PFAS derivatives. Our findings provide critical insights for refining emission control strategies and mitigating the environmental persistence of these compounds." on Page 3

(3) In section 2.1 (Options for electronic structure density functionals), we have added some sentences "... were used to calculate single-point energies. The FNO-CCSD(T) approach that significantly improves computational efficiency with cost reduction of up to an order of magnitude was utilized to calculate larger systems." on Page 3

(4) In section 3.1 (Results and discussion), We have added some sentences for discussion "..., aiming to investigate the effects of increasing carbon chain length on the enthalpy of activation at 0 K. The calculated results show a deviation of only 0.2 kcal/mol in activation enthalpy at 0 K between FNO-CCSD(T)-F12//cc-pVDZ-F12 (-2.6 kcal/mol) and CCSD(T)-F12a/cc-pVTZ-F12 (-2.4 kcal/mol) in CF3CHO + HO2, validating the robustness of FNO-CCSD(T)-F12//cc-pVDZ-F12 for complex fluorinated systems." on Page 7

"For instance, C2F5CHO exhibits three transition state conformers with energy differences spanning 0–1.7 kcal/mol, while C3F7CHO has five conformers distributed over 0–1.8 kcal/mol. This trend amplifies for longer chains: C5F11CHO generates 36 distinct conformers in its transition state, with energy variations extending up to 4.8 kcal/mol." on Page 8

We also added an additional paragraph: "Our further analysis of the global distribution of the ratio between HO2 and OH during daytime reveals that HO2 concentrations are generally higher than those of OH (see Figure S4). Notably, along the west coast of South America (approximately between 0° and 30°S latitude and 60°W to 120°W longitude), the ratio can reach up to three orders of magnitude. Industrial areas (such as Russia and Malaysia) and certain regions in Africa also exhibit high ratios of 1-2 orders of magnitude. This suggests that in these areas, the concentration of HO2 is significantly higher than that of OH, which may be related to local industrial activities or specific emission characteristics. However, due to the presence of daytime photolysis, the generation and loss pathways of HO2 and OH become more complex, leading to significant uncertainty in interpreting the ratio. For instance, photolysis can alter the formation rates of HO2 and OH, thereby affecting their concentration ratio. Additionally, the high ratios along the eastern coast of North America may be associated with atmospheric transport and regional emission features. Despite these complexities, the high daytime ratios still indicate that in specific regions, HO2 may play a role in the oxidation pathways of C2F5CHO and C3H7CHO both during the day and at night. Future research should integrate observational data with model refinements to better quantify the impact of photolysis on the HO2/OH ratio." on Page 14

Figure S4. The annual average ratio of HO2/OH during the day globally. (In Supplement)

(5) In section 4 (Summarizing Remarks), we have revised some sentences from "We construct a comprehensive reaction potential energy surface and find that the activation enthalpies for the reactions of C2F5CHO and C3F7CHO with HO2 at 0 K are both -2.7 kcal/mol, which are exactly the same. Our calculated results suggest that the elongation of the carbon chain in linear perfluorinated aldehyde molecules has a negligible effect on the activation enthalpy." to "We find that the activation enthalpies for the reactions of C2F5CHO and C3F7CHO with HO2 at 0 K are both -2.7 kcal/mol, demonstrating that carbon chain elongation in linear perfluoroaldehydes has a negligible thermodynamic influence on their enthalpies of activation at 0 K. This is further shown in C4F9CHO and C5F11CHO with HO2." on page 15

Revised the sentences from "Additionally, we compared these reactions with the primary oxidation pathway of linear perfluoroaldehydes—their reaction with hydroxyl radicals. We find that there is a big ratio between HO2 and OH concentrations in the Amazon region. The comparative results suggest that the reactions of HO2 with C2F5CHO and C3F7CHO may dominate their atmospheric chemistry in the Amazon region, thereby affecting the environmental impact of these compounds. Based on our estimates, the atmospheric lifetimes of C2F5CHO and C3F7CHO are 14.4-31.3 hours and 21.6-51.8 hours, respectively, and under NO conditions this pathway may be a source of HCOOH and COF2 in the troposphere." to "By integrating kinetics with the data based on GEOS-Chem modelling, we have identified some regions such as Russia, Malaysia, and parts of Africa, where HO2 concentration exceeds OH concentration by 2–3 orders of magnitude. Therefore, the reactions of HO2 with C2F5CHO and C3F7CHO can compete well with their corresponding reaction with OH. Specifically, the atmospheric lifetimes of C2F5CHO and C3F7CHO via HO2 are shortened to be 14.4–31.3 h and 21.6–51.8 h, respectively, with orders of magnitude shorter than that of the corresponding OH-mediated pathways (>20 days). Under

high NOx conditions, this pathway may contribute to tropospheric HCOOH and COF2 formation." on page 15-16

The only new analysis that is available in this work is a GEOS-Chem based atmospheric modelling of the studied reactions to estimate their atmospheric implications. However, that analysis also does not provide any new information that was not known from the work cited above. Similar to CF3CHO, the two larger perfluoroaldehydes studied here also show that reaction with HO2 is more dominant atmospheric removal process compared to reaction with OH.

Therefore, the conclusion that this work "provide new insight into atmospheric degradation of linear perfluorinated aldehydes by HO2 radical" is not supported by the results at all. At most, the study provides new data that shows the atmospheric degradations of larger perfluoroaldehydes by HO2 are very similar to that of CF3CHO which has already been reported earlier.

**Response:** We thank the reviewer for their feedback and agree that the dominance of HO2 as a degradation pathway for linear perfluoroaldehydes aligns with prior observations on smaller analogs like CF3CHO. However, by calculating the HO2-to-OH degradation rate ratios below.

$$v_1 = \frac{k_1[\text{HO}_2]}{k_{\text{OH}}[\text{OH}]}, v_2 = \frac{k_2[\text{HO}_2]}{k'_{\text{OH}}[\text{OH}]}$$

According to the equation above, the rate ratios are largely determined by the ratio of the concentrations of  $HO_2$  and OH. However, their concentrations are varied from one region another region. Geos-Chem analysis can provide further insight into their concentration distribution. For example, in the Amazon, Malaysia, and part of the Africa, elevated  $[HO_2]/[OH]$  ratios are 410–1,200, which lead to the rate ratios of 88.5–259 for  $v_1$  and 56.0–164 for  $v_2$ . In contrast, over oceanic regions like the Atlantic and Pacific, these ratios can drop below 1.

These results are remarkably different from the assumption that the concentrations of HO2 and OH are not dependent on the specific region. Such spatial distribution, driven by localized oxidant, provides novel insights into the atmospheric processing of long-chain PFAS compounds in tropical environments. Thus, while the general dominance of HO2 is acknowledged, our work uniquely quantifies its regional significance, offering new perspectives on the atmospheric degradation of perfluorinated aldehydes.

In order to better present our research results, in the revised manuscript, we modified the sentence from "Especially in the parts of Africa, HO2 is even three orders of magnitude higher than OH. In addition, high concentration ratios have been observed along the Indian Ocean margin near Indonesia, which may be due to atmospheric transport. This large concentration ratios between HO2 and OH suggests that HO2 leads to sink of C2F5CHO and C3H7CHO at night in these particular regions." to "Specifically, in parts of Africa, HO2 concentrations are even three orders of magnitude higher than that of OH. Additionally, high HO2/OH ratios have been observed along the Indian Ocean margin near Indonesia, which may be attributed to atmospheric transport and enhanced HO2 production from industrial activities. In the Amazon region, the [HO2]/[OH] ratio can reach as high as 410-

1,200. This significantly increases the HO2-to-OH degradation rate ratios for C2F5CHO and C3F7CHO, reaching 88.5–259 and 56.0–164, respectively. These rate ratios indicate that HO2-driven degradation exceeds OH-mediated degradation by over 50 times. This large concentration ratios between HO2 and OH suggests that HO2 leads to sink of C2F5CHO and C3H7CHO at night in these regions. In contrast, over oceanic regions like the Atlantic and Pacific, the [HO2]/[OH] ratios drop below 1, leading to a diminished role of HO2 in these areas." on page 13

The most baffling aspect of this work is the sudden introduction of NO into the reaction scheme and an attempt to show the title reaction as a source of formic acid and COF2. However, there is no attempt to calculate the rate constant of the reactions involving NO, which would be required to have any realistic estimate of the importance of NO in determining the atmospheric fate of the studied perfluoroaldehydes. Therefore, the conclusion that "under NO conditions this pathway may be a source of HCOOH and COF2 in the troposphere" is completely unfounded without proper kinetic analysis, including lifetime calculations, of these reaction channels.

**Response:** We appreciate the reviewer's feedback on this point. The inclusion of NO aligns with its established atmospheric relevance as a mediator in radical-driven oxidation cycles, where RO2 + NO reactions are key sinks for peroxy radicals, forming nitrates or terminating chains.(Berndt et al., 2015; King et al., 2001; Nie et al., 2023; Orlando et al., 2000; Vereecken and Peeters, 2009) Our work extends this pattern to perfluoroalkyl systems, demonstrating analogous pathways for M1/M2 (C2F5CH(O)OO/C3F7CH(O)OO) reacting with NO ( $\Delta H_0 = -9.9/-11.5$  kcal/mol at 0 K), reflecting thermodynamic trends in hydrocarbon systems. While detailed kinetics (e.g., rate constants) are essential for quantifying atmospheric impacts, our research focus on identifying thermochemically pathways (e.g., Vereecken & Peeters, 2009, using SAR models to prioritize key channels). The subsequent decomposition of intermediates C2F5CH(O)OH/C3F7CH(O)OH exhibits low barriers (4.7–5.6 kcal/mol at 0 K), suggesting rapid dissociation under tropospheric conditions. However, as emphasized in Section 3.3, the lack of experimental rate constants for perfluoroalkyl-oxy systems precludes definitive quantification of HCOOH/COF2 yields, highlighting the need for future kinetic validation. We have revised the manuscript to clarify the reviewer's concern. Specifically, we have modified the following sections:

3.1. The electronic structure of the C2F5CHO/C3F7CHO + HO2 reaction: "As depicted in Figure S2, M1 and M2 undergo initial reactions with NO to yield the products C2F5CH(O)OH, C3F7CH(O)OH, and NO2, exhibiting activation enthalpies of –9.9 and – 11.5 kcal/mol at 0 K, respectively. These results are consistent with previous studies on similar reactions involving RO2 + NO. (Berndt et al., 2015; King et al., 2001; Nie et al., 2023; Orlando et al., 2000; Vereecken and Peeters, 2009) These products then undergo unimolecular reactions to decompose into C2F5 and C3F7 radicals and formic acid in Figure 2. Notably, the unimolecular decomposition of C2F5CH(O)OH and C3F7CH(O)OH represents the rate-determining step of the overall reaction, with corresponding activation enthalpies of 5.6 kcal/mol and 4.7 kcal/mol (0 K), respectively; this indicates that formic

acid may potentially be formed via C2F5CHO/C3F7CHO + HO2 in the presence of high concentration NO in the atmosphere." on page 6-7

- 3.3 Atmospheric Implications: "The CF3 further reacts to eventually yield COF2 through a similar reaction pathway. However, the absence of quantified rate constants for these reactions prevents a robust assessment of their global or regional impacts. A comprehensive evaluation of the role of NO would require integrating the kinetics of RO2 + NO reactions (e.g., M1/M2 + NO) into atmospheric models, which is beyond the scope of this study." on page 14-15
- 4. Summarizing Remarks: "Therefore, the reactions of HO2 with C2F5CHO and C3F7CHO can compete well with their corresponding reaction with OH. Specifically, the atmospheric lifetimes of C2F5CHO and C3F7CHO via HO2 are shortened to be 14.4–31.3 h and 21.6–51.8 h, respectively, with orders of magnitude shorter than that of the corresponding OH-mediated pathways (>20 days). Under NO-rich conditions, the reaction pathway involving HO2-initiated oxidation of perfluoroaldehydes may serve as a potential source of HCOOH and COF2 in the troposphere." on page 15-16

**References Added in Revision:**

Berndt, T., Richters, S., Kaethner, R., Voigtländer, J., Stratmann, F., Sipilä, M., Kulmala, M., and Herrmann, H.: Gas-Phase Ozonolysis of Cycloalkenes: Formation of Highly Oxidized RO2 Radicals and Their Reactions with NO, NO2, SO2, and Other RO2 Radicals, Journal of Physical Chemistry A, 119, 10336–10348, https://doi.org/10.1021/acs.jpca.5b07295, 2015.

King, M. D., Canosa-Mas, C. E., and Wayne, R. P.: Gas-phase reactions between RO2 and NO, HO2 or CH3O2: correlations between rate constants and the SOMO energy of the peroxy (RO2) radical, Atmospheric Environment, 35, 2081–2088, https://doi.org/10.1016/S1352-2310(00)00501-X, 2001.

Nie, W., Yan, C., Yang, L., Roldin, P., Liu, Y., Vogel, A. L., Molteni, U., Stolzenburg, D., Finkenzeller, H., Amorim, A., Bianchi, F., Curtius, J., Dada, L., Draper, D. C., Duplissy, J., Hansel, A., He, X. C., Hofbauer, V., Jokinen, T., Kim, C., Lehtipalo, K., Nichman, L., Mauldin, R. L., Makhmutov, V., Mentler, B., Mizelli-Ojdanic, A., Petäjä, T., Quéléver, L. L. J., Schallhart, S., Simon, M., Tauber, C., Tomé, A., Volkamer, R., Wagner, A. C., Wagner, R., Wang, M., Ye, P., Li, H., Huang, W., Qi, X., Lou, S., Liu, T., Chi, X., Dommen, J., Baltensperger, U., El Haddad, I., Kirkby, J., Worsnop, D., Kulmala, M., Donahue, N. M., Ehn, M., and Ding, A.: NO at low concentration can enhance the formation of highly oxygenated biogenic molecules in the atmosphere, Nature Communications, 14, 3347, https://doi.org/10.1038/s41467-023-39066-4, 2023. Orlando, J. J., Iraci, L. T., and Tyndall, G. S.: Chemistry of the cyclopentoxy and cyclohexoxy radicals at subambient temperatures, Journal of Physical Chemistry A, 104, 5072–5079, https://doi.org/10.1021/jp0002648, 2000.

Vereecken, L. and Peeters, J.: Decomposition of substituted alkoxy radicals - Part I: A generalized structure-activity relationship for reaction barrier heights, Physical Chemistry Chemical Physics, 11, 9062–9074, https://doi.org/10.1039/b909712k, 2009.

Response to the community #1 Comments for the manuscript "Reaction between perfluoroaldehydes and hydroperoxy radical in the atmosphere: Reaction mechanisms, reaction kinetics modelling, and atmospheric implications"

**Comment:** Line 95: In addition, multi-structural torsional anharmonicity involving reactant and transition state were all calculated using MS-T method (multi-structural method for torsional anharmonicity. How was this done? Can you please elaborate.

**Response:**

**(a)Theoretical background**

The complete conformational-rotational-vibrational partition functions involving reactant and transition state have been all calculated by using MS-T method (multistructural method for torsional anharmonicity)(Zheng and Truhlar, 2013) and executed through MSTor program package.(Chen et al., 2023) In general, the conformational-rotational-vibrational partition function is calculated as,

$$Q_{con-rovib}^{MS-T,X} = \sum_{j=1}^{J} Q_{rot,j} \exp(-\beta U_j) Q_j^{HO} Z_j \prod_{t=1}^{t} f_{j,\tau}$$

$$\tag{1}$$

where "con" and "rovib" denote conformation and rotation-vibration, respectively, J is the number of distinguishable conformational structures ( $j = 1, 2, \dots, J$ ).  $\beta = 1/k_b$ T and X labels reactants, products or transition states.  $Q_{rot,j}$ ,  $U_j$  and  $Q_j^{HO}$  represent the classical rotational partition of structure j, energy gap relative to the global minimum energy structure, and the normal-mode harmonic oscillator vibration partition function of the  $J^{th}$  structure, respectively.  $Z_j$  is a factor to ensure that the MS-T scheme reaches the correct high-Temperatre limit (within the parameter range of the model), and  $f_{j,\tau}$  is an internal coordinate torsional nonharmonic function that adjusts the harmonic partition function of structure j to account for torsional motion  $\tau$ .

When equation (1) is used for a single-structure (SS) rotation-vibrational partition function of the conformer j (generally, the global minima structure (j = 1)), and if  $Z_j$  and  $f_{j,\tau}$  were set to 1, the partition function  $Q_{con-rovib}^{MS-T,X}$  reduces to the multi-structural local-harmonic (MS-LH) partition function. Thus, we can rewrite it as

$$Q_{rovib,i}^{SS-HO} = Q_{rot,i} \exp(-\beta U_i) Q_{vib,i}^{HO}$$
 (2)

Subsequently, the multistructural torsional anharmonicity factor  $F^{MS-T,X}$  will be defined for reactants and transition states by using equation (1) and (2) as shown below

$$F^{MS-T,X} = Q_{con-rovib}^{MS-T,X} / Q_{rovib,1}^{SS-HO}$$
(3)

also, the corresponding multistructural torsional anharmonicity factor for the reaction

$$F^{MS-T,X} = Q_{Sp}^{MS-T} / Q_R^{MS-T} \tag{4}$$

The reaction rate constant for biomolecular was calculated as

$$k^{MS-T} = \kappa \sigma \frac{k_B T}{h} \frac{Q_{elec}^{\neq} Q_{con-rovib}^{MS-T,\neq}}{N_a Q_{elec}^R Q_{con-rovib}^{MS-T,R}} exp(-\beta V^{\neq})$$
 (5)

where  $k_b$  is Boltzmann's constant, T is the temperature, N is Avogadro's number, h is Planck's constant, and  $\beta$  is  $1/k_b$ T.  $V^{\neq}$  denotes the classic reaction energy barrier exclude ZPE correction.  $Q_{elec}$  represent electronic partition function, while  $\kappa$  and  $\sigma$  denote Eckart tunneling coefficient and the reaction symmetry number, respectively.

**(b) Calculated details**

- (1) Conformational Search: Use **ConfGen.exe** to generate initial structures by rotating specified bonds
- (2) Geometry Optimization & Frequency Calculation: Optimize structures using Gaussian and remove duplicates (e.g., mirrored/rotationally equivalent structures). Run frequency calculations to obtain Hessians and energies.
- (3) Distinct Structure Identification: Use **mvinput.exe** to exclude mirror images and rotationally redundant structures. Generate *mvorm.inp* for Voronoi tessellation
- (4) Torsional Periodicity  $(M_{j,\tau})$ : Compute  $M_{j,\tau}$  values using **mcvorm.exe** (Monte Carlo) or vorm.exe (Voronoi tessellation).
- (5) Input File Generation: Combine Gaussian .fchk files into *all.fchk*. Use **msinput.exe** to generate *mstor.inp* and *hess.dat*, adding uncoupled torsions.
- (6) Execution: Run mstor.exe with the MS-T(CD) method to calculate partition functions and thermodynamic properties.
- (c) Input files for the ConfGen.exe executable

13 2

C 1.01911500 -0.47626600 0.05443400

```
\mathbf{C}
           2.35234900
                       2.51946100
Η
                       0.29297300
                                    1.53563000
C
                       0.51749400 0.11794900
           -0.15660500
O
           3.12769000
                       0.56284200 -0.38553200
F
                       1.46045400 -0.81804700
           0.04052700
F
          -0.13406100
                       1.10278400 1.33631400
F
           1.10175100 -0.98403000 -1.18192700
F
           0.78274500 -1.47995300 0.93436300
\mathbf{C}
           -1.56560300 -0.07991200 -0.09014200
F
          -2.45510800
                       0.91017100 -0.11075900
F
          -1.62361100 -0.73826800 -1.24442900
F
          -1.87185600 -0.91027500 0.90264400
**torsion 1 definition**
2 1
3
523
3
0.0 120.0 -120.0
**torsion 2 definition**
14
6
189235
3
0.0 120.0 -120.0
%nprocshared=12
%mem=80GB
**M062x/gen opt=calcfc int=grid=99974 scf=conver=11**
0 1
@/home/zgdong/mg3s.gbs
Extbasis
```

**Reference**

Chen, W., Zheng, J., Bao, J. L., Truhlar, D. G., and Xu, X.: MSTor 2023: A new version of the computer code for multistructural torsional anharmonicity, now with automatic torsional identification using redundant internal coordinates, Computer Physics Communications, 288, 108740, https://doi.org/10.1016/j.cpc.2023.108740, 2023.

Zheng, J. and Truhlar, D. G.: Quantum thermochemistry: Multistructural method with torsional anharmonicity based on a coupled torsional potential, Journal of Chemical Theory and Computation, 9, 1356–1367, https://doi.org/10.1021/ct3010722, 2013.

---

## Author Response (AR2)

Response to the Anonymous Referee #2 Comments for the manuscript "Reaction between perfluoroaldehydes and hydroperoxy radical in the atmosphere: Reaction mechanisms, reaction kinetics modelling, and atmospheric implications"

**Comments:**

The authors have now provided their response to the comments I made based on their original submission and has modified the manuscript accordingly. They have provided three key points to describe the effect of chain elongation on reaction thermodynamics and kinetics:

- 1. Longer chain leads to multiple conformers (for both reactants and TSs) resulting in reduction in the multi-structure torsional anharmonicity factor and consequently decrease in rate constants by 50 % at most.
- 2. C3F7CHO shows a significantly higher pressure dependence compared to its two smaller counterparts. However, they do not provide any explanation for this anomalous behavior.
- 3. They develop a computational strategy based on the frozen natural orbital (FNO) approximation for the longer (or larger) compounds that provides computational efficiency.

The authors have carried out an extensive work and the work deserves publication. However, the information is incremental in nature (like, decrement in rate constant, increasing pressure dependence in one single case) and does not provide any new insight (like, change in mechanistic pathway, new atmospheric loss channels, significant change in rate constants by orders of magnitude, systematic modulation in temperature or pressure dependence with structural changes etc.) that warrants publication in atmospheric chemistry and physics. Their development of a new computational strategy to efficiently deal with bigger molecules makes this work suitable for computational chemistry related journals. Therefore, I do not recommend publication of this manuscript in atmospheric chemistry and physics based on the above comments.

**Response:** We agree with your views that the original article does not provide further insight into the degradation of linear perfluoroaldehydes. Therefore, in the revised article, we have done additional computations and provided a new and full analysis on the atmospheric degradation of linear perfluoroaldehydes, containing homogeneous and heterogeneous processes. We have rewritten Section 3.3 in the revised manuscript, with corresponding minor refinements incorporated into the Abstract and Conclusions to ensure consistency

**Atmospheric Implications:** "To provide a further insight into the atmospheric degradation pathways of linear perfluoroaldehydes, we compare the HO2-initiated linear perfluoroaldehyde reactions with the corresponding reactions with OH and Cl atom, their photolysis and hydrolysis.

We quantitatively evaluate the relative importance of OH- versus HO2-initiated degradation for

C2F5CHO and C3F7CHO through rate ratios defined in equations (5) and (6).

$$v_{1} = \frac{k_{1}[C_{2}F_{5}CHO][HO_{2}]}{k_{OH}[C_{2}F_{5}CHO][OH]}$$

$$v_{2} = \frac{k_{2}[C_{3}F_{7}CHO][HO_{2}]}{k'_{OH}[C_{3}F_{7}CHO][OH]}$$
(6)

Here,  $k_1$  and  $k_2$  are the rate constants of HO2 + C2F5CHO and HO2 + C3F7CHO calculated in the present work, respectively, while  $k_{OH}$  and  $k'_{OH}$  are the corresponding rate constants of OH+ C2F5CHO and OH + C3F7CHO obtained in the literature.(Solignac et al., 2007b; Wang et al., 2007) We calculate the rate ratios using a high OH concentration of  $5 \times 10^6$  molecules cm-3 (Lew et al., 2020) and a typical HO2 concentration of  $1.4 \times 10^8$  molecules cm-3 (Brasseur and Solomon, 2006). The calculated results reveal that within the temperature range of 220-320 K, the rate ratios for  $v_1$  and  $v_2$  are in the range of Table 1. Rate ratios of HO2 + C2F5CHO to OH + C2F5CHO and HO2 + C3F7CHO to OH + C3F7CHO within the

Table 1. Rate ratios of HO2 + C2F5CHO to OH + C2F5CHO and HO2 + C3F7CHO to OH + C3F7CHO within the Temperature Range of 240 to 350 K.

| T(K)             | $k_1{}^a$              | $\frac{k_2'^a}{}$      | $v_1{}^b$         | $v_2^b$ |
|------------------|------------------------|------------------------|-------------------|---------|
| 220              | $9.04 \times 10^{-13}$ | $6.10 \times 10^{-13}$ | <mark>79.2</mark> | 45.7    |
| 240       | $4.64 \times 10^{-13}$ | $3.12 \times 10^{-13}$ | 34.22             | 20.34   |
| <mark>260</mark> | $2.68 \times 10^{-13}$ | $1.79 \times 10^{-13}$ | 17.09             | 10.38   |
| 280              | $1.70 \times 10^{-13}$ | $1.13 \times 10^{-13}$ | 9.55              | 5.90    |
| <mark>298</mark> | $1.19 \times 10^{-13}$ | $7.92 \times 10^{-13}$ | 6.06              | 3.83    |
| 320              | $8.21 \times 10^{-13}$ | $5.48 \times 10^{-13}$ | 3.76              | 2.43    |

 $^{a}k_{1}$  and  $k'_{2}$  are the rate constants of the HO2 reactions with C2F5CHO and C3F7CHO, from the literature respectively.  $^{b}v_{1}$  and  $v_{2}$  denote the rate ratios of HO2 with C2F5CHO and C3F7CHO to OH with C2F5CHO and C3F7CHO, respectively.

79.2 to 3.76 and 45.7 to 2.43, respectively. Therefore, the present findings indicate that HO2 initiated reactions dominate over OH initiated reactions for the degradation of C2F5CHO and C3F7CHO. We further consider the atmospheric lifetimes of C2F5CHO and C3F7CHO with respect to HO2 at 0–50 km altitude in Table 2. Rapid HO2-initiated degradation leads to short atmospheric lifetimes of ~14.4–31.3 hours for C2F5CHO and 21.6–51.8 hours for C3F7CHO (Table 2), which are significantly shorter than the ~20-day atmospheric lifetime driven by OH oxidation at below 10 km.(Antiñolo et al., 2014)

**Table 2.** Hydroperoxyl radical concentration (in molecules cm-3), rate constants (in cm3 molecule-1 s -1), and atmospheric lifetimes (in hours) with respect to bimolecular reactions as functions of altitude.

| H (km) a | $T(K)^a$           | P (mbar) a | $[HO_2]^b$           | $k_1(T, p)^{c}$        | $k_2(T,p)^c$           | ${f 	au_1}^{f d}$ | ${	au_2}^{ m d}$     |
|---------------------|--------------------|-----------------------|----------------------|------------------------|------------------------|-------------------|----------------------|
| 0                   | <mark>290.2</mark> | 1010                  | $1.40 \times 10^{8}$ | $1.38 \times 10^{-13}$ | $9.18 \times 10^{-14}$ | 14.4              | 21.6                 |
| 5            | 250.5       | 496                   | $4.90 \times 10^{7}$ | $3.44 \times 10^{-13}$ | $2.30 \times 10^{-13}$ | 16.5              | 24.7                 |
| 10                  | 215.6              | 243                   | $8.30 \times 10^{6}$ | $1.07 \times 10^{-12}$ | $6.47 \times 10^{-13}$ | 31.3              | 51.8                 |
| 15                  | 198                | 119                   | $2.30 \times 10^{6}$ | $2.21 \times 10^{-12}$ | $2.98 \times 10^{-13}$ | <del>54.7</del>   | $4.05 \times 10^{2}$ |
| <mark>20</mark>     | <mark>208</mark>   | 58.2           | $2.90 \times 10^{6}$ | $1.38 \times 10^{-12}$ | $1.22 \times 10^{-13}$ | 69.3              | $7.85 \times 10^{2}$ |
| <mark>25</mark>     | 216.1       | 28.5                  | $5.70 \times 10^{6}$ | $9.38 \times 10^{-13}$ | $4.32 \times 10^{-14}$ | 52.0              | $1.13 \times 10^3$   |
| <mark>30</mark>     | 221.5              | 13.9                  | $7.50 \times 10^{6}$ | $6.52 \times 10^{-13}$ | $1.29 \times 10^{-14}$ | <mark>56.8</mark> | $2.88 \times 10^{3}$ |

| 35       | 228.1              | 6.83  | $6.90 \times 10^6$   | $4.04 \times 10^{-13}$ | $4.13 \times 10^{-15}$ | <mark>99.6</mark>    | $9.74 \times 10^{3}$ |
|-----------------|--------------------|-------|----------------------|------------------------|------------------------|----------------------|----------------------|
| <mark>40</mark> | 240.5              | 3.34  | $5.90 \times 10^6$   | $2.29 \times 10^{-13}$ | $1.73 \times 10^{-15}$ | $2.06 \times 10^{2}$ | $2.73 \times 10^4$   |
| <mark>45</mark> | <mark>251.9</mark> | 1.64  | $4.90 \times 10^{6}$ | $1.27 \times 10^{-13}$ | $6.56 \times 10^{-16}$ | $4.45 \times 10^{2}$ | $8.64 \times 10^{4}$ |
| <mark>50</mark> | 253.7       | 0.801 | $4.00 \times 10^{6}$ | $5.87 \times 10^{-13}$ | $1.80 \times 10^{-16}$ | $1.18 \times 10^{3}$ | $3.86 \times 10^{5}$ |

aH denotes altitude (atmospheric scale height); T denotes temperature; p denotes pressure. bData are from ref (Brasseur and Solomon, 2006).  $^{c}k_{1}$ ,  $k_{2}$  are the rate constants of the HO2 reactions with C2F5CHO and C3F7CHO, respectively.  $^{d}\tau_{1} = 1/(k_{1}[HO_{2}])$  and  $\tau_{2} = 1/(k_{2}[HO_{2}])$  define the atmospheric lifetimes for HO2 reactions with C2F5CHO and C3F7CHO, respectively.

To provide further insight into the degradation of C2F3CHO and C3F7CHO under atmospheric conditions, further analysis has been done based on Geos-Chem data. GEOS-Chem simulations indicate that HO2 concentrations reach a maximum of 4.99 × 108 molecules cm-3 in the Amazon region, with a mean value of 9.93 × 107 molecules cm-3. (Long et al., 2024) In contrast, the maximum OH concentration over the Atlantic and Pacific oceans is found to be 8.03 × 106 molecules cm-3, with an average value of 1.06 × 106 molecules cm-3. (Lelieveld et al., 2016) However, the OH concentration remarkably differ from daytime to nighttime. (Bey et al., 1997; Stone et al., 2012) Therefore, we consider the concentration ratio between HO2 and OH during the nighttime and daytime. During the nighttime, as shown in Fig. 9, the concentration ratio of [HO2]/[OH] exceeds two orders of magnitude in industrial regions such as Russia, Malaysia, and parts of Africa, with values reaching as high as 410–1,200 in the Amazon. This markedly enhances the HO2-to-OH degradation rate ratios for C2F3CHO and C3F7CHO, reaching values of 88.5–259 and 56.0–164, respectively. This suggest that HO2-initiated degradation exceeds OH-initiated pathways by more than a factor of 50 during nighttime in these regions. In contrast, over oceanic regions such as the Atlantic and Pacific, the [HO2]/[OH] ratio falls below unity, substantially reducing the contribution of HO2 to degradation processes in these areas.

Figure 9. The annual average ratio of HO2/OH at night globally.

During the nighttime, as shown in Figure S4, the concentration ratios between HO2 and OH generally favor HO2, exhibiting maxima up to three orders of magnitude along the western coast of South America and ranging from one to two orders of magnitude in industrialized and African regions. These elevated ratios are closely associated with localized emission sources. However, During the daytime, photolysis is also an important route for removal of C2F5CHO and C3F7CHO. For C2F5CHO and C3F7CHO, photolysis represents the dominant atmospheric degradation route. C2F5CHO exhibits a high photolysis quantum yield of 0.81 ± 0.09 at 254 nm, corresponding to an estimated atmospheric lifetime of less than two days in Table 3. Similarly, C3F7CHO displays a measured photolysis lifetime of 21 ± 10 hours under sunlight, confirming the efficiency of this removal mechanism.(Chiappero et al., 2006)(Solignac et al., 2007a).

Chlorine atom reactions represent an additional potential atmospheric degradation pathway for linear perfluorinated aldehydes. Kinetic measurements report rate constants of  $k(\text{Cl} + \text{C}_2\text{F}_5\text{CHO}) = (1.96 \pm 0.28) \times 10^{-12} \text{ cm}^3 \text{ molecule}^{-1} \text{ s}^{-1} \text{ (Sulbaek Andersen et al., 2003)}$  and  $k(\text{Cl} + \text{C}_3\text{F}_7\text{CHO}) = (2.03 \pm 0.23) \times 10^{-12} \text{ cm}^3 \text{ molecule}^{-1} \text{ s}^{-1} \text{ (Andersen et al., 2004)}$ , which are approximately one order of magnitude higher than those for the corresponding HO2-initiated reactions. Although Cl atoms exhibit intrinsically faster reaction kinetics, the atmospheric relevance of this degradation pathway is limited by the relatively low concentrations of Cl in the troposphere. Typical Cl atom concentrations range from  $1.0 \times 10^4$  to  $3.0 \times 10^5$  molecules cm-3 (Chang et al., 2004; Hossaini et al., 2016; Wang et al.,

2019), leading to estimated atmospheric lifetimes of approximately 400 hours (~17 days) for both C2F5CHO and C3F7CHO. This stands in sharp contrast to the significantly shorter HO2-driven lifetimes, which are typically less than 79.2 hours in the lower troposphere in Table 3. The predominance of HO2-mediated degradation arises from the relatively high ambient concentrations of HO2, which compensate for its slower reaction kinetics—particularly in regions such as the Amazon, where [HO2]/[Cl] ratios may exceed ~103.(Li et al., 2018; Wang et al., 2019) Under such conditions, the HO2 intiated reaction rate can exceed that of Cl by 2–3 orders of magnitude. Moreover, the atmospheric relevance of Cl-initiated degradation is further constrained by its spatial heterogeneity, being primarily restricted to marine boundary layers and polluted coastal environments.(Hossaini et al., 2016; Yang et al., 2022) In contrast, HO2-driven degradation is effective across continental interiors and industrialized regions.

Table 3. Atmospheric lifetimes (τ, hours) of C2F5CHO and C3F7CHO against major degradation pathways.

| Reactant/Process                | C                      | C 2 F 5 CHO | C 3 F 7 CHO |                          |  |
|---------------------------------|------------------------|-----------------------------------|-----------------------------------|--------------------------|--|
| Reactant/110ccss                | τ (hours)              | Ref.                              | τ (hours)                         | Ref.                     |  |
| $HO_2$                          | 3.76-79.2              | This work                         | 2.43-45.7                         | This work                |  |
| ОН                              | 90.8-174.2             | (Antiñolo et al., 2014)           | 79.7-113.0                        | (Solignac et al., 2007b) |  |
| Photolysis                      | <48                    | (Chiappero et al., 2006)          | 14.6-39.7                         | (Solignac et al., 2007b) |  |
| Cl                              | 424.4-571.6            | (Sulbaek Andersen et al., 2003)   | 409.7-514.4                       | (Andersen et al., 2004)  |  |
| Hydrolysis a         | $>5.39 \times 10^6$    | This work                         |                            |                   |  |
| HCOOH
catalysis b | $>5.06 \times 10^7$    | This work                         | >1.13 × 10 8           | This work                |  |
| Heterogeneous
hydrolysis °   | 3.47× 10 -4 | This work                         | 1                                 | 4                        |  |

aGas-phase hydrolysis of C₂F₅CHO with H₂O. bHCOOH-catalyzed gas-phase hydrolysis of C₂F₅CHO/C₃F⁊CHO with H₂O. cHydrolysis of C₂F₅CHO with water dimer at the air-water interface.

In addition to photolysis and radical-initiated oxidation, hydrolysis constitutes another potential atmospheric sink for C2F5CHO and C3F7CHO. Taking C2F5CHO hydrolysis as an example, its gas phase hydrolysis proceeds extremely slowly, with an estimated atmospheric lifetime exceeding 5.39 × 106 hours (Table 3 and Table S11). This removal pathway is negligible, aligning with findings reported for CF3CHO.(Sulbaek Andersen et al., 2006) Hydrolysis catalyzed by atmospheric acids could potentially enhance hydrolysis rates through its ability to reduce the reaction barriers.(Hazra et al., 2013; Liu et al.,

2021) Formic acid (HCOOH), a ubiquitous atmospheric component, forms stable complexes with water (HCOOH····H2O). Even at elevated concentrations of HCOOH····H2O complexes (e.g., 1011 molecule cm-3), the estimated hydrolysis lifetimes of C2F5CHO and C3F7CHO exceed 107 hours in Table 3. These timescales remain orders of magnitude longer than those associated with HO2-initiated degradation, suggesting that the acid-catalyzed hydrolysis is insufficient to promote significant atmospheric removal of these compounds. Therefore, although acid catalysis effectively reduces the reaction barrier, its impact on the gas-phase degradation of C2F5CHO and C3F7CHO is negligible under typical tropospheric conditions.

Hydrolysis at air-water interfaces, such as those present on aerosol particles and cloud droplets, proceeds with markedly enhanced efficiency. Laboratory experiments have shown that passing gaseous CF3CHO through liquid water results in over 80% conversion to CF3CH(OH)2 within seconds, highlighting the potential importance of heterogeneous processes in atmospheric removal pathways.(Sulbaek Andersen et al., 2006) Similarly, 1H NMR measurements reveal that C2F5CHO rapidly converts to its gem-diol form, CF3CF2CH(OH)2, within 3 minutes upon contact with liquid water, further confirming the efficient aqueous-phase hydration of perfluorinated aldehydes. (Sulbaek Andersen et al., 2006) Here, we estimate a low Gibbs free-energy barrier (ΔG = 9.8 kcal/mol) for C2F5CHO + 3H2O at air-water interfaces, proceeding via a cyclic proton-transfer mechanism by using ab initio molecular dynamics, compared to a much higher barrier of 25.5 kcal/mol for the corresponding gas-phase reaction (See Figure S5a, b). More details are provided in Supplementary Material. This results in significantly shorter atmospheric lifetimes, on the order of  $3.47 \times 10^{-4}$  hours. Under humid conditions, such air—water interfacial hydrolysis is likely to dominate and may effectively compete with HO2-mediated degradation pathways. Once formed, CnF2n+1CH(OH)2 reacts with OH, ultimately leading to the formation of perfluorocarboxylic acids (PFCAs). This hydrolysis-oxidation pathway represents a significant indirect source of persistent PFCAs, particularly given the ubiquity of aqueous phases in the atmosphere.

We can conclude that  $HO_2$  intiated degradation pathways dominate the gas-phase degradation of  $C_2F_5CHO$  and  $C_3F_7CHO$ . We further consider the final product in the  $HO_2 + C_nF_{2n}CHO$  reactions. As mentioned, the  $HO_2$  reaction generates intermediate perfluoroalkyl radicals  $C_nF_{2n+1}$ , which can subsequently undergo a carbon-shortening process to form the more stable  $COF_2$ , as depicted in Figure 10. Taking the example of  $C_2F_5$ , the process starts with  $C_2F_5$  reacting with  $O_2$  to form  $C_2F_5O_2$ . Subsequently,  $C_2F_5O_2$  reacts with NO to produce  $C_2F_5O$ , which then undergoes C-C bond cleavage to

generate CF3 and COF2. The CF3 further reacts to eventually yield COF2 through a similar reaction pathway. However, the absence of quantified rate constants for these reactions prevents a robust assessment of their global or regional impacts. A comprehensive evaluation of the role of NO would require integrating the kinetics of RO2 + NO reactions (e.g., M1/M2 + NO) into atmospheric models, which is beyond the scope of this study.

Figure 10. Atmospheric degradation mechanism for  $C_nF_{2n+1}$  with  $C_2F_5$  used as a representative example.

In summary, we identify that the HO2-initiated reaction represents an important atmospheric sink for linear perfluoroaldehydes in gas phase. Notably, recent studies suggest that HO2 concentrations may be elevated at air—water interfaces compared to the bulk gas phase. (Angelaki et al., 2024; Li et al., 2023) Given the enhanced reactivity at the air—water interface and the complex competition between OH and HO2, interfacial HO2-driven degradation may play a more significant role than previously recognized, potentially influencing atmospheric acidity. For example, Xia et al. (2024) recently reported both single-carbon and double-carbon scission pathways during the degradation of C7F15 on water droplet surfaces. These heterogeneous processes may contribute not only to the atmospheric removal of perfluoroaldehydes but also to the broader degradation of polyfluoroalkyl substances (PFAS). Nevertheless, further experimental and theoretical studies on reaction kinetics and mechanisms are needed to better constrain this complex chemical processes. Incorporating such processes into atmospheric models is crucial for improving the prediction of PFAS environmental fate and secondary pollution, with important implications for emission control strategies and environmental risk assessment." (page 12-17, lines 270-383)

**Abstract:** "...pressures ranging from 0.026 to 2.3 bar across a temperature range of 190–350 K. Furthermore, atmospheric lifetimes of C2F5CHO and C3F7CHO are discussed based on the homogenous and heterogeneous processes. Our findings also reveal that the reactions of C2F5CHO and C3F7CHO with HO2 radicals dominate over those with OH radicals in Russia, Malaysia, parts of Africa by the calculated

results in combination with data based on global atmospheric chemical model simulations. HO2-initiated degradation represents a major atmospheric sink, compared to photolysis and Cl-initiated oxidation in gas phase at night, whereas hydrolysis at air-water interface plays a critical role in the sink of linear perfluoroaldehydes." (page 1, lines 16-17, and 19-21)

Summarizing Remarks: "...shorter than that of the corresponding OH-mediated pathways. In addition, photolysis, typically occurring within 48 hours, represents an efficient daytime removal pathway, while heterogeneous hydrolysis proceeds rapidly at the air-water interfaces with characteristic timescales of less than 1 hour. Accordingly, HO2-initiated degradation should be considered a major gas-phase sink, particularly in continental source regions." (page 17, lines 398-401)

**References (in revised manuscript)**

Andersen, M. P. S., Nielsen, O. J., Hurley, M. D., Ball, J. C., Wallington, T. J., Stevens, J. E., Martin, J. W., Ellis, D. A., and Mabury, S. A.: Atmospheric chemistry of n- $C_xF_{2x+1}$ CHO (x = 1, 3, 4): Reaction with Cl atoms, OH radicals and IR spectra of  $C_xF_{2x+1}$ C(O)O2NO2, Journal of Physical Chemistry A, 108, 5189–5196, https://doi.org/10.1021/jp0496598, 2004.

Angelaki, M., Carreira Mendes Da Silva, Y., Perrier, S., and George, C.: Quantification and Mechanistic Investigation of the Spontaneous  $H_2O_2$  Generation at the Interfaces of Salt-Containing Aqueous Droplets, Journal of the American Chemical Society, 146, 8327–8334, https://doi.org/10.1021/jacs.3c14040, 2024. Antiñolo, M., Jiménez, E., González, S., and Albaladejo, J.: Atmospheric chemistry of  $CF_3CF_2CHO$ : Absorption cross sections in the UV and IR regions, photolysis at 308 nm, and gas-phase reaction with OH radicals (T = 263-358 K), Journal of Physical Chemistry A, 118, 178–186, https://doi.org/10.1021/jp410283v, 2014.

Bey, I., Aumont, B., and Toupance, G.: The nighttime production of OH radicals in the continental troposphere, Geophysical Research Letters, 24, 1067–1070, https://doi.org/10.1029/97GL00889, 1997.

Brasseur, G. and Solomon, S.: Aeronomy of the middle atmosphere: chemistry and physics of the stratosphere and mesosphere., Springer Science & Business Media, 617–627 pp., https://doi.org/10.1007/1-4020-3824-0, 2006.

Chang, C. T., Liu, T. H., and Jeng, F. T.: Atmospheric concentrations of the Cl atom, CIO radical, and HO radical in the coastal marine boundary layer, Environmental Research, 94, 67–74,

https://doi.org/10.1016/j.envres.2003.07.008, 2004.

Chiappero, M. S., Malanca, F. E., Argüello, G. A., Wooldridge, S. T., Hurley, M. D., Ball, J. C., Wallington, T. J., Waterland, R. L., and Buck, R. C.: Atmospheric chemistry of perfluoroaldehydes (CxF2x+1CHO) and fluorotelomer aldehydes (CxF2x+1CH2CHO): Quantification of the important role, of photolysis, Journal of Physical Chemistry A, 110, 11944–11953, https://doi.org/10.1021/jp064262k, 2006.

Hazra, M. K., Francisco, J. S., and Sinha, A.: Gas phase hydrolysis of formaldehyde to form methanediol: Impact of formic acid catalysis, Journal of Physical Chemistry A, 117, 11704–11710, https://doi.org/10.1021/jp4008043, 2013.

Hossaini, R., Chipperfield, M. P., Saiz-Lopez, A., Fernandez, R., Monks, S., Feng, W., Brauer, P., and Von Glasow, R.: A global model of tropospheric chlorine chemistry: Organic versus inorganic sources and impact on methane oxidation, Journal of Geophysical Research, 121, 14271–14297, https://doi.org/10.1002/2016JD025756, 2016.

Lelieveld, J., Gromov, S., Pozzer, A., and Taraborrelli, D.: Global tropospheric hydroxyl distribution, budget and reactivity, Atmospheric Chemistry and Physics, 16, 12477–12493, https://doi.org/10.5194/acp-16-12477-2016, 2016.

Lew, M. M., Rickly, P. S., Bottorff, B. P., Reidy, E., Sklaveniti, S., Léonardis, T., Locoge, N., Dusanter, S., Kundu, S., Wood, E., and Stevens, P. S.: OH and HO2 radical chemistry in a midlatitude forest: Measurements and model comparisons, Atmospheric Chemistry and Physics, 20, 9209–9230, https://doi.org/10.5194/acp-20-9209-2020, 2020.

Li, K., Guo, Y., Nizkorodov, S. A., Rudich, Y., Angelaki, M., Wang, X., An, T., Perrier, S., and George, C.: Spontaneous dark formation of OH radicals at the interface of aqueous atmospheric droplets, Proceedings of the National Academy of Sciences of the United States of America, 120, e2220228120, https://doi.org/10.1073/pnas.2220228120, 2023.

Li, M., Karu, E., Brenninkmeijer, C., Fischer, H., Lelieveld, J., and Williams, J.: Tropospheric OH and stratospheric OH and Cl concentrations determined from CH4, CH3Cl, and SF6 measurements, npj Climate and Atmospheric Science, 1, 29, https://doi.org/10.1038/s41612-018-0041-9, 2018.

Liu, J. Y., Long, Z. W., Mitchell, E., and Long, B.: New Mechanistic Pathways for the Reactions of Formaldehyde with Formic Acid Catalyzed by Sulfuric Acid and Formaldehyde with Sulfuric Acid Catalyzed by Formic Acid: Formation of Potential Secondary Organic Aerosol Precursors, ACS Earth

**and Space Chemistry, 5, 1363-1372, https://doi.org/10.1021/acsearthspacechem.1c00002, 2021.**

Long, B., Zhang, Y.-Q., Xie, C., Tan, X.-F., and Truhlar, D. G.: Reaction of Carbonyl Oxide with Hydroperoxymethyl Thioformate: Quantitative Kinetics and Atmospheric Implications, Research, 7, 525, https://doi.org/10.34133/research.0525, 2024.

Solignac, G., Mellouki, A., Le Bras, G., Yujing, M., Sidebottom, H., Goulay, F., Osborn, D. L., Taatjes, C. A., Zou, P., Meloni, G., and Leone, S. R.: The gas phase tropospheric removal of fluoroaldehydes  $(C_xF_{2x+1}CHO, x = 3, 4, 6)$ , Physical Chemistry Chemical Physics, 9, 4200–4210, https://doi.org/10.1039/B703741B, 2007a.

Solignac, G., Mellouki, A., Le Bras, G., Yujing, M., and Sidebottom, H.: The gas phase tropospheric removal of fluoroaldehydes ( $C_xF_{2x+1}CHO$ , x =3,4,6), Physical Chemistry Chemical Physics, 9, 4200–4210, https://doi.org/10.1039/b614502g, 2007b.

Stone, D., Whalley, L. K., and Heard, D. E.: Tropospheric OH and HO2 radicals: Field measurements and model comparisons, Chemical Society Reviews, 41, 6348–6404, https://doi.org/10.1039/c2cs35140d, 2012.

Sulback Andersen, M. P., Hurley, M. D., Wallington, T. J., Ball, J. C., Martin, J. W., Ellis, D. A., Mabury, S. A., and Nielsen, O. J.: Atmospheric chemistry of C2F5CHO: Reaction with Cl atoms and OH radicals, IR spectrum of C2F5C(O)O2NO2, Chemical Physics Letters, 379, 28–36, https://doi.org/10.1016/j.cplett.2003.08.004, 2003.

Sulback Andersen, M. P., Toft, A., Nielsen, O. J., Hurley, M. D., Wallington, T. J., Chishima, H., Tonokura, K., Mabury, S. A., Martin, J. W., and Ellis, D. A.: Atmospheric Chemistry of Perfluorinated Aldehyde Hydrates ( $n-C_xF_{2x+1}CH(OH)_2$ , x=1,3,4): Hydration, Dehydration, and Kinetics and Mechanism of Cl Atom and OH Radical Initiated Oxidation, The Journal of Physical Chemistry A, 110, 9854–9860, https://doi.org/10.1021/jp060404z, 2006.

Wang, X., Jacob, D. J., Eastham, S. D., Sulprizio, M. P., Zhu, L., Chen, Q., Alexander, B., Sherwen, T., Evans, M. J., Lee, B. H., Haskins, J. D., Lopez-Hilfiker, F. D., Thornton, J. A., Huey, G. L., and Liao, H.: The role of chlorine in global tropospheric chemistry, Atmospheric Chemistry and Physics, 19, 3981–4003, https://doi.org/10.5194/acp-19-3981-2019, 2019.

Wang, Y., Liu, J. yao, Yang, L., Zhao, X. lei, Ji, Y. M., and Li, Z. sheng: Theoretical studies and rate constant calculations of the reactions C2F5CHO with OH radicals and Cl atoms, Journal of Molecular Structure: THEOCHEM, 820, 26–34, https://doi.org/10.1016/j.theochem.2007.06.001, 2007.

Xia, D., Zhang, H., Ju, Y., Xie, H., Su, L., Ma, F., Jiang, J., Chen, J., and Francisco, J. S.: Spontaneous Degradation of the "Forever Chemicals" Perfluoroalkyl and Polyfluoroalkyl Substances (PFASs) on Water Droplet Surfaces, Journal of the American Chemical Society, 146, 11266–11271, https://doi.org/10.1021/jacs.4c00435, 2024.

Yang, X., Wang, Q., Ma, N., Hu, W., Gao, Y., Huang, Z., Zheng, J., Yuan, B., Yang, N., Tao, J., Hong, J., Cheng, Y., and Su, H.: The impact of chlorine chemistry combined with heterogeneous N2O5 reactions on air quality in China, Atmospheric Chemistry and Physics, 22, 3743–3762, https://doi.org/10.5194/acp-22-3743-2022, 2022.